# ON THE ANISOTROPY OF SCORE-BASED GENERATIVE MODELS

## ABSTRACT

We investigate the role of network architecture in shaping the inductive biases of modern score-based generative models. To this end, we introduce the *Score Anisotropy Directions* (SADs), architecture-dependent directions that reveal how different networks preferentially capture data structure. Our analysis shows that SADs form adaptive bases aligned with the architecture's output geometry, providing a principled way to predict generalization ability in score models prior to training. Through both synthetic data and standard image benchmarks, we demonstrate that SADs reliably capture fine-grained model behavior and correlate with downstream performance, as measured by Wasserstein metrics. Our work offers a new lens for explaining and predicting directional biases of generative models.[1]

## 1 INTRODUCTION

Neural networks generalize through inductive biases, i.e., biases that guide learning beyond training data (Goyal & Bengio, 2020; Wilson & Izmailov, 2020). For discriminative tasks, they are partially characterized through the *Neural Anisotropy Directions* (NADs), which reveal the architecture's directional preferences in the input space (Ortiz-Jimenez et al., 2020). However, generative modeling lacks a cohesive theory that explains how architectural geometry interacts with data manifolds (Kadkhodaie et al., 2024; An et al., 2025). In this work, we present a unified approach to explaining and interpreting inductive biases of score-based generative models by examining anisotropy in the output space, where networks exhibit preferential learning along certain directions. As

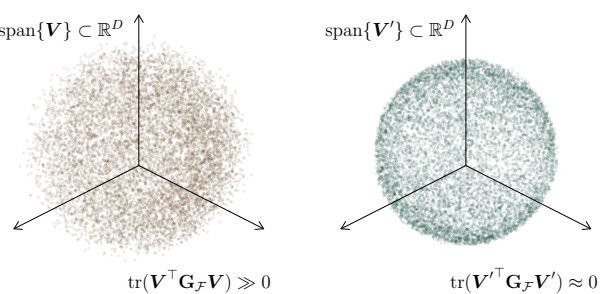

Figure 1: Sphere modeling in subspaces of $\mathbb{R}^D$ ($D = 256$) via DiT (Peebles & Xie, 2023). The only difference is the choice of subspace: the left "sphere" lies in a subspace aligned with the network's geometry, $\mathbf{G}_\mathcal{F}$, while the right is in a non-aligned subspace. Despite identical setups, their quality differs consistently across repeated trials, suggesting that alignment with architectural geometry controls generalization. We formalize these ideas in Sections 3.2 and 3.3.

motivated in Figure 1, we posit that generalization ability is largely characterized by the alignment of the data with the architecture's "geometry" at initialization. Our contribution is making this notion of "geometry" precise and decomposing the output space in terms of the anisotropy directions induced by it, i.e., the *Score Anisotropy Directions* (SADs) that we introduce below:

**Definition 1** (**Score Anisotropy Directions**). The *Score Anisotropy Directions* (SADs) of an architecture are the ordered set of orthonormal vectors of the output space, $\{\boldsymbol{u}_i\}_{i=1}^D$, that are ranked in terms of the preference (e.g., as measured via Wasserstein metrics) of the network to generate data along those particular directions via score-based generative modeling.

---

[1]Code to reproduce our experiments is included in the supplementary material.

## 2 BACKGROUND

Here we give a brief overview of score-based generative models and provide some context on existing work that examines inductive biases of deep neural networks. We defer an extended discussion and comparison with related work to Section 4.1.

### 2.1 SCORE-BASED GENERATIVE MODELS

At the heart of score-based modeling is the *score function*, i.e., the gradient of the log-density of the underlying data distribution $p$. Given an estimate of the score, one can, in theory, sample from $p$ via a gradient ascent procedure. Starting with an arbitrary prior, $\boldsymbol{x}_0 \sim \pi$, we have the following Langevin dynamics iterations:

$$\boldsymbol{x}_{k+1} = \boldsymbol{x}_k + \frac{\eta}{2} \overbrace{\nabla_{\boldsymbol{x}_k} \log p(\boldsymbol{x}_k)}^{\text{"score"}} + \sqrt{\eta} \boldsymbol{z}_k, \quad k = 0, 1, \dots, K \tag{1}$$

where $\boldsymbol{z}_k \sim \mathcal{N}(\boldsymbol{0}, \boldsymbol{I})$ is standard Gaussian and $\eta > 0$. If $\eta \to 0$ and $K \to \infty$, under certain technical conditions, the iterates in Equation 1 converge to a sample from $p$ (Welling & Teh, 2011). However, a practical limitation of the above setup is that estimated scores are inaccurate in low-density regions (e.g., in early iterations where learning is intractable). Moreover, score functions may be undefined in the case of data residing in low-dimensional manifolds. That is, the overall approach breaks down under the commonly adopted manifold hypothesis (Song & Ermon, 2019).

The research community has therefore largely moved on to Denoising Score Matching (DSM) (Vincent, 2011) and annealed Langevin dynamics, i.e., diffusion models, which are also the main focus of this work.[2] Specifically, consider noise scales, $\sigma \in [\sigma_{\min}, \sigma_{\max}]$, and associated densities, $p_\sigma = p * \mathcal{N}(\boldsymbol{0}, \sigma^2 \boldsymbol{I})$, where $*$ represents convolution. Here, $\sigma_{\min}$ is small enough such that $p_{\sigma_{\min}} \approx p$ and $\sigma_{\max}$ is large enough so we can write $p_{\sigma_{\max}} \approx \mathcal{N}(\boldsymbol{0}, \sigma_{\max}^2 \boldsymbol{I})$. With estimates of scores of the perturbed distributions, $\nabla_{\boldsymbol{x}_\sigma} \log p_\sigma(\boldsymbol{x}_\sigma)$, one samples from $p$ by decaying $\sigma$ from $\sigma_{\max}$ to $\sigma_{\min}$ (Ho et al., 2020; Song et al., 2021; Miyato et al., 2025; Song et al., 2025). This way, accurate modeling along sampling trajectories is tractable and the noise ensures support over the entirety of the ambient space, overcoming the limitations of naive Langevin dynamics. In particular, the scores are equivalent to minimum mean squared error Gaussian denoisers (Efron, 2011). That is, for neural networks, $\mathcal{F}_{\boldsymbol{\theta}} : \mathbb{R}^D \times \mathbb{R} \to \mathbb{R}^D$, parameterized by $\boldsymbol{\theta}$, one can approximate the scores via the following DSM optimization:

$$\min_{\boldsymbol{\theta}} \mathbb{E}_{\boldsymbol{x} \sim p, \boldsymbol{\epsilon} \sim \mathcal{N}(\boldsymbol{0}, \boldsymbol{I}), \sigma} \left[ \hat{\mathcal{J}}_{\text{DSM}} := \left\| \mathcal{F}_{\boldsymbol{\theta}}(\boldsymbol{x} + \sigma \boldsymbol{\epsilon}, \sigma) + \frac{\boldsymbol{\epsilon}}{\sigma} \right\|_2^2 \right]. \tag{2}$$

### 2.2 INDUCTIVE BIASES IN DEEP LEARNING

There is a vast literature on understanding the inductive biases of deep neural networks (Goyal & Bengio, 2020; Wilson & Izmailov, 2020; Radhakrishnan et al., 2024). Of particular interest is the work of Ortiz-Jimenez et al. (2020), who identify directional biases in classifiers, i.e., the *Neural Anisotropy Directions* (NADs). More recently, Movahedi et al. (2025) have extended the NAD framework by formalizing the concept of input-space architectural geometry and exploring its evolution during training. Specifically, they propose the *geometric invariance hypothesis*, which posits that NADs persist through training.

To our knowledge, contrary to the discriminative case, there is no unified theory that explains the preferred modeling directions of diffusion. For example, a recent study by Kadkhodaie et al. (2024) argues that convolutional diffusion models are biased towards *Geometry-Adaptive Harmonic Bases* (GAHBs). However, they acknowledge that a mathematically precise definition of such bases remains an open question. More fundamentally, follow-up work by An et al. (2025) suggests that the theory of GAHBs does not extend to transformer-based diffusion models.

---

[2] Without loss of generality, we will standardize notation to the variance exploding formulation of Song & Ermon (2019).

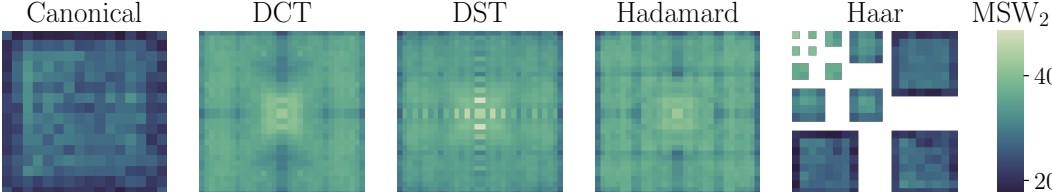

Figure 2: $\text{MSW}_2$ distance (computed over 10k test samples and 16384 projections) of iDDPM U-Net (Nichol & Dhariwal, 2021) architecture. Each pixel corresponds to a rank-one dataset of 16×16 images (with 10k training samples) that is aligned with a basis element of the canonical basis, DCT, DST, (ordered) Hadamard transform or Haar wavelet transform. That is, for a given location (canonical) or frequency / sequency (DCT, DST, Hadamard) or scale, channel and location (Haar), we visualize the performance on the corresponding dataset. For ease of visualization, in the case of DCT, DST and Hadamard, we center the zero frequency dataset and extend the images to the left and top regions while respecting the symmetries of the transforms. See Appendix A for details.

In contrast to existing works on inductive biases of diffusion (Kadkhodaie et al., 2024; An et al., 2025), we present a unified treatment of directional biases that accounts for network architecture. In particular, our key insight is recognizing that the NAD framework can be adapted and extended to score-based modeling by assuming an underlying data log-density that assigns high probability to "on-manifold" data and low probability otherwise. With this approach, our analysis amounts to understanding the anisotropy directions of such implicitly induced discriminative models.

## 3 DIRECTIONAL INDUCTIVE BIASES OF DIFFUSION MODELS

**Notation** Matrix entries are indexed as $(\cdot)_{(i,j)}$ for row $i$ and column $j$. Unless otherwise specified, we write the $i^{\text{th}}$ row of a matrix as $(\cdot)^{(i)}$. $\otimes$ is the Kronecker product. $\delta_{(\cdot)}$ denotes the Kronecker delta or Dirac delta distribution, depending on the context. The term iid refers to independent and identically distributed random variables. Gaussian distributions with mean $\boldsymbol{\mu}$ and covariance $\boldsymbol{\Sigma}$ are represented as $\mathcal{N}(\boldsymbol{\mu}, \boldsymbol{\Sigma})$. $\mathcal{U}(S)$ is the uniform distribution over the set $S$. We abbreviate (Stochastic) Gradient Descent as (S)GD. Eigendecompositions of matrices are assumed to have the eigenvalues in descending order, so that the first eigenvector corresponds to the largest eigenvalue. We assume that all expectations appearing in our arguments are finite and well-defined.

Central to our exploration of directional inductive biases in diffusion is the following question:

*Among equidimensional manifolds, which are preferred by diffusion-based generative modeling and how can such preferences be quantified?*

To answer the above, we consider data manifolds aligned with a particular direction, $\boldsymbol{v} \in \mathbb{S}^{D-1}$, and investigate generalization ability as a function of the direction. Concretely, we study distributions of the form $\mathcal{N}(\mathbf{0}, D\boldsymbol{v}\boldsymbol{v}^\top)$ and are tasked with finding a suitable basis for $\mathbb{R}^D$ from which we can draw $\boldsymbol{v}$ that reveal directional preferences. We include complete details of our experimental setup in Appendix A. In summary, for a given basis, we independently train diffusion models on datasets formed by each normalized element under identical settings. Performance is quantified via the (Max-)Sliced Wasserstein $p$-distance, (M)SW$_p$, between the distribution obtained by sampling from the trained model, $\mu$, and the ground truth data, corresponding to $\nu = \mathcal{N}(\mathbf{0}, D\boldsymbol{v}\boldsymbol{v}^\top)$. Notably, these are valid statistical distances and the slice distance, $\text{W}_2$, is easily estimated via order statistics and Monte Carlo methods. Taking $p = 2$ and letting $F^{-1}_{(\cdot)}$, $(\cdot)_\#$ represent quantile functions, the push-forward respectively, we express the Wasserstein metrics theoretically as:

$$\text{SW}_2^2(\mu, \nu) = \mathbb{E}_{\boldsymbol{\theta} \sim \mathcal{U}(\mathbb{S}^{D-1})} \text{W}_2^2(\boldsymbol{\theta}_\#^\top \mu, \boldsymbol{\theta}_\#^\top \nu), \quad \text{MSW}_2^2(\mu, \nu) = \sup_{\boldsymbol{\theta} \in \mathbb{S}^{D-1}} \text{W}_2^2(\boldsymbol{\theta}_\#^\top \mu, \boldsymbol{\theta}_\#^\top \nu),$$

$$\text{W}_2^2(\boldsymbol{\theta}_\#^\top \mu, \boldsymbol{\theta}_\#^\top \nu) = \int_0^1 |F^{-1}_{\boldsymbol{\theta}_\#^\top \mu}(q) - F^{-1}_{\boldsymbol{\theta}_\#^\top \nu}(q)|^2 \mathrm{d}q.$$

(3)

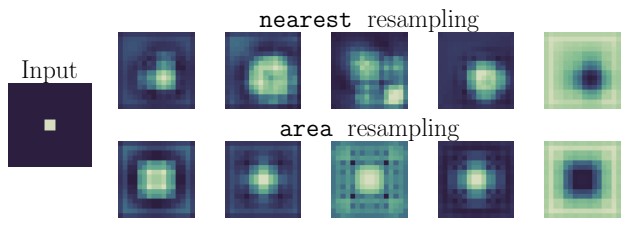

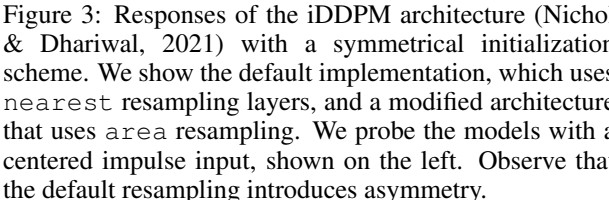

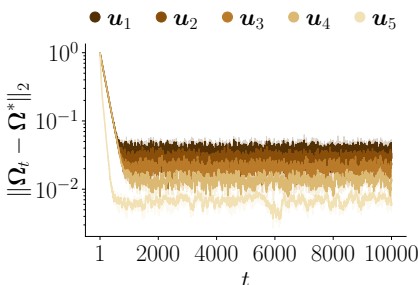

Figure 3: Responses of the iDDPM architecture (Nichol & Dhariwal, 2021) with a symmetrical initialization scheme. We show the default implementation, which uses `nearest` resampling layers, and a modified architecture that uses `area` resampling. We probe the models with a centered impulse input, shown on the left. Observe that the default resampling introduces asymmetry.

Figure 4: Setting of Theorem 1 ($\sigma = 1$) in $\mathbb{R}^5$ with SGD. We show error between $\mathbf{\Omega}_t = \mathbf{\Phi}\mathbf{\Theta}_t$ and the optimal operator, $\mathbf{\Omega}^*$, defined in Lemma 1.

In Figure 2 we report our findings (averages over five runs) with the above-described approach for common bases found in signal processing literature. Interestingly, contrary to the conventional wisdom that neural networks better adapt to lower frequencies (Rahaman et al., 2019), the standard U-Net used in DDPMs (Nichol & Dhariwal, 2021) struggles with low-frequency data. This is evidenced by the center points of the DCT, DST, Hadamard images and the top-left corner of the Haar image. Also, comparing results of the canonical and Haar bases with the frequency / sequency transforms, we see that vectors localized in space are better modeled, especially around the borders.

### 3.1 ANISOTROPIC CONDITIONING OF THE OPTIMIZATION LANDSCAPE

Why would certain directions be preferred? While the learning process has a number of hyperparameters that can potentially induce asymmetry, here we specifically focus on analyzing the role of the architecture. Prior work on the inductive biases of discriminative models argues that preferences may emerge due to anisotropic loss of information or, more generally, conditioning of the optimization landscape (Ortiz-Jimenez et al., 2020). Indeed, such conditioning also naturally manifests in multiscale U-Nets, for example, in the form of asymmetric resampling layers. We verify this in Figure 3, where we observe that the default `nearest` interpolation leads to a clear directional bias. Similarly, one may hypothesize that the border effects observed in the canonical and Haar basis experiments of Figure 2 can be attributed to the standard zero padding strategy employed in the convolutional layers. Also, at the same time, one might intuitively expect that data aligned with such padding artifacts is poorly approximated by the network.

To more rigorously understand the effect of anisotropic conditioning, we now revisit the problem of learning rank-one distributions, introduced in the beginning of Section 3, from a theoretical perspective. Assuming a linear DSM architecture, where anisotropy is explicitly modeled via a fixed, linear transformation with decaying eigenvalues at the output, we make the following observation:

**Theorem 1** (DSM under anisotropy, proof in Appendix D.2). *Consider the DSM problem with data drawn from $\mathcal{N}(\mathbf{0}, \mathbf{v}\mathbf{v}^\top)$ for a fixed noise level $\sigma > 0$ and $\|\mathbf{v}\|_2 = 1$. Let $\mathcal{F} : \mathbb{R}^D \to \mathbb{R}^D$ be linear networks expressed as $\mathbf{\Omega}(\cdot)$, where $\mathbf{\Omega} = \mathbf{\Phi}\mathbf{\Theta}$ with $\mathbf{\Phi}$ fixed. Denote the sorted eigenvalues of $\mathbf{\Phi}\mathbf{\Phi}^\top$ as $\{\lambda_i\}_{i=1}^D$ with $\lambda_{D-1} > \lambda_D > 0$ and the corresponding normalized eigenvectors as $\{\mathbf{u}_i\}_{i=1}^D$. Assume a (S)GD procedure on initially zero-mean $\mathbf{\Theta}$, where the score is approximated by $\mathcal{F}$. After $t$ steps, and with sufficiently small learning rate $\eta > 0$, the mean error, $\mathbb{E}[\mathbf{\Omega} - \mathbf{\Omega}_*]$, to the optimal solution, $\mathbf{\Omega}_*$, for $\mathbf{v} = \mathbf{u}_i$ decays as $\mathcal{O}[(1 - 2\eta\rho_i)^t]$ with $\rho_1 = \rho_i < \rho_D \; \forall i < D$. Moreover, near optimality, the SGD steps with respect to $\mathbf{\Theta}$ for $\mathbf{v} = \mathbf{u}_i$ have covariance $\mathrm{Cov}[\nabla_\mathbf{\Theta} \hat{\mathcal{J}}_{\mathrm{DSM}}] \propto \lambda_i$.*

Curiously, the effect of anisotropy is limited in deterministic GD, i.e., if $\lambda_D = \lambda_{D-1}$ then it completely disappears and the convergence rate is the same for all eigenvectors of $\mathbf{\Phi}\mathbf{\Phi}^\top$. However, our derivations suggest that anisotropy is greatly amplified in SGD. To confirm this, we design a small-scale experiment in $\mathbb{R}^5$, with the results shown in Figure 4. Specifically, for small $t$, the optimization dynamics are well-predicted by the deterministic GD analysis of Theorem 1 since, intuitively, the iterates are far from the optimum and the deterministic drift, i.e., the gradient, dominates the stochastic fluctuation. Consequently, all vectors except $\mathbf{u}_5$ yield similar results. For large $t$, however, where

the gradient norm is sufficiently small, the noise dominates and it appears we have convergence to a stationary distribution. In such stochastic regimes, the error scales with the magnitude of the stochastic gradient covariance (Mandt et al., 2017), which we show to be $\propto \lambda_i$ near optimality.

The takeaway from this discussion is that the best performance is achieved when the data is *not* aligned with the "geometry" that is induced by the score network, but instead lives in the subspace defined by its *smallest* eigenvalues. Next, we will extend and formalize this intuition in a more general setting, for potentially non-linear architectures.

### 3.2 Identifying Preferred Directions for Generative Modeling

Although the experiments of Figure 2 are insightful, it is unclear whether standard bases can reliably describe the biases of diffusion models given that they are completely decoupled from the underlying architecture. As the choice of bases was arbitrary in these initial experiments, we argue that, in general, we cannot hope to uncover the intricacies of directional biases in this manner. Moreover, note that each considered basis amounts to training hundreds of diffusion models, which becomes impractical even for relatively small-dimensional data. We now present a more principled approach to these experiments, by establishing theoretically motivated and architecture-dependent bases that predict directional biases prior to any training.

Concretely, for each noise level, $\sigma$, and assuming a (conservative and normalizable) parameterization via a neural network family, $\mathcal{F}$, we intuitively treat a realization, $\mathcal{F}_{\boldsymbol{\theta}} : \mathbb{R}^D \times \mathbb{R} \to \mathbb{R}$, as implicitly defining a log-density function, $\boldsymbol{x}_\sigma \mapsto \log p_{\boldsymbol{\theta},\sigma}(\boldsymbol{x}_\sigma)$, that assigns high probability to $\boldsymbol{x}_\sigma$ on the "data manifold" and low probability to "off-manifold" data, i.e., we can write $\mathcal{F}_{\boldsymbol{\theta}}(\boldsymbol{x}_\sigma, \sigma) \approx \nabla_{\boldsymbol{x}_\sigma} \log p_{\boldsymbol{\theta},\sigma}(\boldsymbol{x}_\sigma)$. Here, we use the term "manifold" loosely, meaning regions of $\mathbb{R}^D$ that are statistically likely to be sampled by score-based modeling via $\mathcal{F}_{\boldsymbol{\theta}}$, i.e., the spanning set of the top SADs that we aim to uncover. Now, intuitively, if one were to fix $\boldsymbol{x}_\sigma$ and consider a small perturbation along some direction, $\boldsymbol{v}$, an abrupt change in the log-density indicates falling off or entering the "manifold", that is, $\boldsymbol{v} = \boldsymbol{v}_\perp$ is perpendicular to the "manifold". Similarly, if $\boldsymbol{v} = \boldsymbol{v}_\parallel$ is parallel to the "manifold", then we expect minimal changes in the log-density, i.e., we are traveling along a contour line. We illustrate our argument in Figure 5. Now, taking the limit as the perturbation magnitude, $\tau$, tends to zero, we summarize these dynamics via directional derivatives,

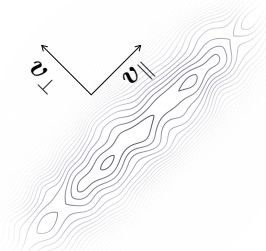

Figure 5: Visualization of our argument for uncovering anisotropy directions in $\mathbb{R}^2$. We show contours of a hypothetical landscape, $\log p_{\boldsymbol{\theta},\sigma}(\boldsymbol{x}_\sigma)$, where $\boldsymbol{v}_\parallel$ is parallel to the induced "manifold" and $\boldsymbol{v}_\perp$ is orthogonal.

i.e., we have $|\log p_{\boldsymbol{\theta},\sigma}(\boldsymbol{x}_\sigma + \tau \boldsymbol{v}) - \log p_{\boldsymbol{\theta},\sigma}(\boldsymbol{x}_\sigma)| \propto |\boldsymbol{v}^\top \nabla_{\boldsymbol{x}_\sigma} \log p_{\boldsymbol{\theta},\sigma}(\boldsymbol{x}_\sigma)|$. With this simplification, a straightforward application of Markov's inequality bounds the *a priori* probability of crossing the "manifold" by moving along $\boldsymbol{v}$, suggesting that directions attaining the minimum upper bound are inherently easier to model. That is, such directions are aligned with the prior density induced by $\mathcal{F}_{\boldsymbol{\theta}}$. Specifically, for the family of networks, $\mathcal{F}$, parameterized by $\boldsymbol{\theta} \sim \Theta$ over noise levels, $\sigma$, and probing with $(\boldsymbol{x}_\sigma, \sigma) \sim \mathcal{P}$, we write the bound:

$$\mathbb{P}\big(\big|\boldsymbol{v}^\top \nabla_{\boldsymbol{x}_\sigma} \log p_{\boldsymbol{\theta},\sigma}(\boldsymbol{x}_\sigma)\big| \geq \eta\big) \leq \frac{\boldsymbol{v}^\top \overbrace{\Big[\mathbb{E}_{(\boldsymbol{x}_\sigma,\sigma)\sim\mathcal{P},\,\boldsymbol{\theta}\sim\Theta}\mathcal{F}_{\boldsymbol{\theta}}(\boldsymbol{x}_\sigma,\sigma)\,\mathcal{F}_{\boldsymbol{\theta}}(\boldsymbol{x}_\sigma,\sigma)^\top\Big]}^{\text{``geometry''}} \boldsymbol{v}}{\eta^2}. \quad (4)$$

We note that by applying this bound on the setting of Theorem 1 with $\boldsymbol{\theta}$ iid and for any $\mathcal{P}$, the quantity under the overbrace, namely the *average geometry*, recovers the conditioning matrix, $\boldsymbol{\Phi}\boldsymbol{\Phi}^\top$, whose eigendecomposition defined the SADs in the case of linear networks. Moreover, the Markov bound correctly predicts that the small-eigenvalue vectors are easier to model compared to larger-eigenvalue eigenvectors. We therefore expect that this quantity also captures directional preferences of more general architectures. As the average geometry is central to our study of directional biases, let us formally introduce it below.

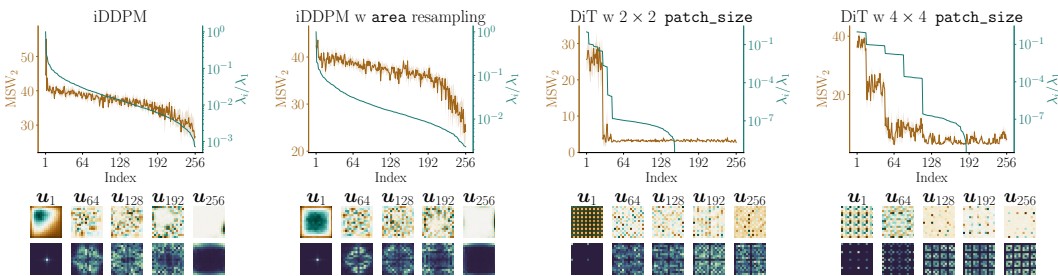

Figure 6: Test $MSW_2$ distance for different architectures on datasets aligned with the eigenvectors of their geometry at initialization, probing with $\mathcal{P} = \delta_0 \times \mathcal{U}(\{\sigma_{\min}, \ldots, \sigma_{\max}\})$. We report the mean $\pm$ the standard error over five independent runs. Corresponding normalized eigenvalues are on the right axes. The eigenvectors, with their energy in the DFT (zero frequency is centered), are shown below the plots (first, last row respectively). The experimental setup is identical to the one described in the beginning of Section 3 and Figure 2. We refer the reader to Appendix A for further details.

**Definition 2** (**Average Geometry**). Let $\mathcal{F}$ be a family of neural networks parameterized by $\boldsymbol{\theta} \sim \Theta$ (e.g., at initialization) such that $\mathbb{R}^D \times \mathbb{R} \to \mathbb{R}^D$. In the context of diffusion-based generative modeling, we define the average geometry of $\mathcal{F}$, induced by a probing distribution, $\mathcal{P}$, and parameterized by $\Theta$, as:

$$\mathbf{G}_{\mathcal{F}}(\mathcal{P}, \Theta) = \mathbb{E}_{(\boldsymbol{x}_\sigma, \sigma) \sim \mathcal{P}, \boldsymbol{\theta} \sim \Theta} \left[ \mathcal{F}_{\boldsymbol{\theta}}(\boldsymbol{x}_\sigma, \sigma) \mathcal{F}_{\boldsymbol{\theta}}(\boldsymbol{x}_\sigma, \sigma)^\top \right], \tag{5}$$

where we assume the networks are, roughly, aligned with some underlying data density, $p_{\boldsymbol{\theta}, \sigma}(\boldsymbol{x}_\sigma)$, via $\mathcal{F}_{\boldsymbol{\theta}}(\boldsymbol{x}_\sigma, \sigma) \approx \nabla_{\boldsymbol{x}_\sigma} \log p_{\boldsymbol{\theta}, \sigma}(\boldsymbol{x}_\sigma)$. Note, the probing distribution, $\mathcal{P}$, is chosen independently and need not match this underlying density.

At this point it is important to note that, unlike in the linear networks investigated in Theorem 1, the notion of average geometry is, in general, local and adapts to the probing data distribution $\mathcal{P}$. To decouple the geometry from the data, one may consider an isotropic probe, e.g., $\mathcal{N}(\boldsymbol{0}, \sigma_{\mathcal{P}}^2 \boldsymbol{I})$. However, in practice, we find that the probe is a non-critical hyperparameter and not central to our analysis. For simplicity, we default to $\mathcal{P} = \delta_0 \times \mathcal{U}(\{\sigma_{\min}, \ldots, \sigma_{\max}\})$, i.e., we fix $\boldsymbol{x}_\sigma = \boldsymbol{0}$ and draw $\sigma$ uniformly from the noise levels of interest. We shall write the geometry under this default probe as $\mathbf{G}_{\mathcal{F}}$, where the parameters, $\boldsymbol{\theta}$, of $\mathcal{F}$ are assumed to be drawn from the default initialization scheme for the architecture. Having formalized the concept of output-space geometry, we are now ready to state our main conjecture regarding the SADs, linking them to the average geometry at initialization:

*Conjecture* 1. Let $\mathcal{F}$ be a family of networks with geometry $\mathbf{G}_{\mathcal{F}}$. We hypothesize that the eigenvectors of $\mathbf{G}_{\mathcal{F}}$, in *ascending* eigenvalue order, are the SADs. That is, we expect that data aligned with eigenvectors corresponding to small eigenvalues is better modeled compared to data that is aligned with large-eigenvalue eigenvectors.

In the remainder of the paper, we provide empirical evidence to justify Conjecture 1. First, we verify our claims on the rank-one datasets introduced in the beginning of Section 3 in Figure 6, where we draw the directions, $\boldsymbol{v} \in \mathbb{R}^D$, from the eigenvectors of $\mathbf{G}_{\mathcal{F}}$ and benchmark via Wasserstein metrics.

On the left in Figure 6, we focus on the iDDPM U-Net (Nichol & Dhariwal, 2021), which is representative of convolutional diffusion models. The experiments show a clear trend in support of the conjecture, where eigenvectors corresponding to small eigenvalues achieve the best performance and large-eigenvalue vectors have the worst performance. Interestingly, with reference to the visualizations below the plots, we also observe harmonic patterns in the eigenvectors, where large eigenvalues correspond to low-frequencies and small eigenvalues to high-frequencies. In this sense, our findings provide further evidence in support of the theory of GAHBs (Kadkhodaie et al., 2024), which posits that harmonic representations are fundamental in convolutional models.[3] Crucially, beyond the harmonic structure that only loosely characterizes the inductive biases, our notion of geometry is flexible as it captures and adapts to architectural details. This is evident in the first eigenvectors, $\boldsymbol{u}_1$, when we ablate the resampling method, also investigated previously in Figure 3.

---

[3]See Figure 9 in Appendix B for more visualizations that connect our work with Kadkhodaie et al. (2024).

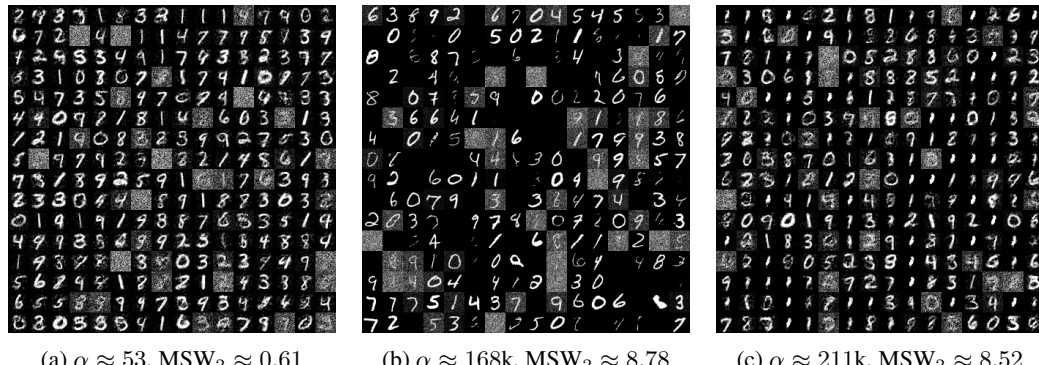

(a) $\alpha \approx 53$, $MSW_2 \approx 0.61$      (b) $\alpha \approx 168k$, $MSW_2 \approx 8.78$      (c) $\alpha \approx 211k$, $MSW_2 \approx 8.52$

Figure 7: Uncurated samples of MNIST-trained iDDPMs. We vary the alignment of the data with the geometry, $\alpha := \mathbb{E}_{\boldsymbol{x} \sim p}[\boldsymbol{z}^\top \mathbf{G}_{\mathcal{F}} \boldsymbol{z}]$, by transforming the data via appropriate orthogonal matrices $\boldsymbol{W}$. On the left, we show the effect of minimizing $\alpha$, where the model gives a reasonable approximation of the ground truth distribution. The middle shows the default alignment, i.e., we do not apply any transformation. It is evident that the data is not well-modeled in this case as a considerable fraction of samples do not contain a digit, as also quantified via the $MSW_2$. The right shows samples from the model corresponding to maximizing $\alpha$, where we see similar $\alpha$ and $MSW_2$ to the middle. We also observe signs of mode collapse, where the model is more likely to generate the digit "1".

As further proof of the flexibility of our proposed average geometry, we now focus our analysis on the DiT architecture (Peebles & Xie, 2023), which is representative of transformer-based diffusion models. The results, shown on the right in Figure 6, are also in support of Conjecture 1. However, as also noted in (An et al., 2025), it is clear in this case that the transformer architecture does not induce harmonic representations and the theory of GAHBs does not apply. It is also interesting to observe that the DiT geometry exhibits considerable eigen-multiplicity, with larger patch sizes amplifying this, reflecting a looser structure and weaker inductive biases compared to convolutional networks. In fact, in Proposition 3 we show that the number of transformer tokens, $T$, is an upper bound on the number of distinct eigenvalues of $\mathbf{G}_{\mathcal{F}}$. More generally, we refer the reader to Appendix C for an analytical treatment of geometries of common architectures, noting that a surprising amount of structure can be inferred simply by inspecting the output layers.

### 3.3 Score Anisotropy Directions in the Wild

Having identified the preferred modeling directions in our experiments on rank-one datasets, we now turn to more realistic data distributions that are encountered in practice. Based on our analysis in Section 3.2, we hypothesize that generalization is largely determined by the (mis)alignment of the data with the average geometry. For an arbitrary data distribution, $p$, we can extend our setup by defining the alignment with the network, $\alpha$, as follows:

$$\alpha := \mathbb{E}_{\boldsymbol{x} \sim p}[\boldsymbol{z}^\top \mathbf{G}_{\mathcal{F}} \boldsymbol{z}], \quad \boldsymbol{z} = \boldsymbol{W}\boldsymbol{x}, \quad \boldsymbol{W}^\top \boldsymbol{W} = \boldsymbol{I}. \tag{6}$$

Note, in order to vary $\alpha$ in a way that preserves underlying structure, we have introduced the orthogonal matrix $\boldsymbol{W} \in \mathbb{R}^{D \times D}$, which models simple and lossless data transformations, effectively defining a linear autoencoder and a kind of "latent" diffusion model that operates on the transformed data. With this setup, Conjecture 1 amounts to the claim that the best performance is observed when $\alpha$ is minimized and the worst when $\alpha$ is maximized. The corresponding orthogonal matrices, $\boldsymbol{W}_{\min}$ and $\boldsymbol{W}_{\max}$, that achieve such extreme (mis)alignment are given by Theorem 2:

**Theorem 2** (Extreme alignment, proof in Appendix D.3). *Stacking the (descending by eigenvalue magnitude) eigenvectors of $\mathbf{G}_{\mathcal{F}}$ and $\mathbb{E}_{\boldsymbol{x} \sim p}[\boldsymbol{x}\boldsymbol{x}^\top]$ as columns in matrices $\boldsymbol{U}, \boldsymbol{V} \in \mathbb{R}^{D \times D}$ respectively, $\alpha$ is minimized for $\boldsymbol{W}_{\min} = \boldsymbol{U}\boldsymbol{R}\boldsymbol{V}^\top$ and maximized for $\boldsymbol{W}_{\max} = \boldsymbol{U}\boldsymbol{V}^\top$. Here, $\boldsymbol{R}$ is the row-reversed identity matrix, i.e., it reverses the order of the columns in $\boldsymbol{V}$.*

Therefore, to test our hypothesis, for each $W$, we will train such "latent" diffusion models under identical settings. To compare with the standard diffusion training, we also consider a baseline corresponding to $W = I$, i.e., the natural alignment of the data with the average geometry. In particular, we conduct experiments on the MNIST (LeCun et al., 2010) ($28 \times 28$), CelebA-HQ (Karras et al., 2018) ($56 \times 56$) and CIFAR-10 (Krizhevsky, 2009) ($32 \times 32$) datasets, with implementation details included in Appendix A. Our findings, shown in Figures 7 and 8, agree with our central conjecture. For example, focusing on the MNIST samples obtained by our models, which are shown in Figure 7, we observe that large $\alpha$ results in artifacts on the distribution-level.[4] Specifically for the default alignment, which is already significant, a considerable fraction of the samples do not contain any digit. Moreover, maximizing $\alpha$ leads to mode collapse, with the digit "1" being overrepresented. When we instead minimize $\alpha$, substantial and consistent improvements are observed across all datasets, as quantified by the Wasserstein

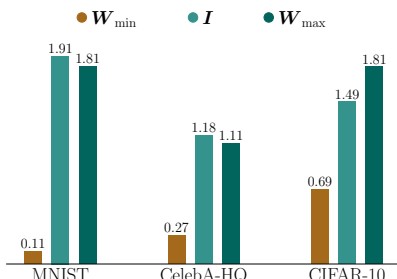

Figure 8: iDDPM average performance ($SW_2$) over five independent runs on standard image datasets as the alignment with the geometry, $\alpha$, varies. We show the effect of minimizing, maximizing $\alpha$ as well as the default setting. These experiments correspond to matrices $W_{\min}$, $W_{\max}$ and $I$ respectively.

metrics in Figure 8. Interestingly, we find that, by default, the considered datasets are already well-aligned with $\mathbf{G}_{\mathcal{F}}$. Indeed, our analysis shows that explicitly maximizing $\alpha$ also yields very similar scores to the baseline $W = I$, which is consistent with our hypothesis that the optimization dynamics of diffusion models are tightly linked to the interaction of data and architecture via $\alpha$. In particular, since both $W = W_{\max}$ and $W = I$ exhibit high alignment, their performance differences may not be statistically significant. The more meaningful comparison is with $W = W_{\min}$, where alignment is minimal and the Wasserstein metrics are also substantially lower in all cases.

## 4 DISCUSSION

We have presented and experimentally validated a theory for determining the directional inductive biases of diffusion models. Quite surprisingly, we find that, despite the highly non-linear nature of modern neural networks, these biases are well-described via fixed bases, that may be decoupled from the data and only depend on the architecture, making them useful for predicting generalization ability, as quantified via Wasserstein metrics. Specifically, we have shown that the directions defined by these bases impose strong priors during the training process and define the order by which features of the data are learned, i.e., we identify them as the SADs.

### 4.1 RELATED WORK

Kadkhodaie et al. (2024) observe that convolutional diffusion models are inductively biased toward GAHBs. Specifically, by assuming a homogeneous model (Mohan et al., 2020), they give a shrinkage-based interpretation of denoising with adaptive eigenbases given by the local Jacobian of the trained networks. A similar Jacobian-based analysis has also been explored in transformer-based diffusion (An et al., 2025). However, the observations of Kadkhodaie et al. (2024) are architecture-specific and largely empirical, therefore they cannot be used to predict preferred modeling directions in general. For example, transformer-based diffusion exhibits no obvious regularities in the eigenbases, such as harmonic structure, that could be systematically exploited in An et al. (2025). Despite using a fundamentally different approach, our theory of SADs appears to extend and is compatible with the findings of these works. In particular, we provide a more rigorous treatment of architectural geometry, decoupling it from the data and we demonstrate a method that can predict generalization ability *prior to any training*. We achieve this by decomposing and, crucially, *ordering the output space* via the SADs, which are defined explicitly. This ordering provides fine-grained information about inductive biases, rather than a loose characterization. Moreover, unlike prior Jacobian-based analyses, we make no assumptions regarding homogeneity, which may be suboptimal / not hold across domains or architectures, and our notion of geometry only requires forward passes through

---

[4]Note that, in general, we do not expect perceptual quality to correlate with the Wasserstein distances.

the networks. This makes our approach more broadly applicable and straightforward to implement. As future work, we remark that it remains an open question whether our method has theoretical ties with the analysis initially proposed by Kadkhodaie et al. (2024).

We note that, with some variations, a notion similar to our average geometry also appeared previously in Ortiz-Jimenez et al. (2020); Movahedi et al. (2025), who study the directional inductive biases of discriminative networks. Interestingly, their analysis predicts that classifiers actually perform better for data aligned with the *largest* eigenvalues of the geometry. Though their setup is not identical to ours, and therefore not directly comparable, we believe that these seemingly contrasting conclusions may hint at a fundamental trade-off between discriminative and generative modeling. However, such an investigation is out of the scope of this paper and we leave it as future work.

Given prior work that connects the NADs with Neural Tangent Kernel (NTK) theory (Jacot et al., 2018; Ortiz-Jimenez et al., 2021), another interesting direction for future research is to investigate potential links between the SADs and the NTK. While SADs capture architectural anisotropy by exploiting the structure of score functions, relying only on forward passes and implicit modeling, the NTK directly uses network Jacobians to describe optimization dynamics of wide neural networks. Regardless of the connection with the NTK, the present findings are interesting on their own as they provide insights on finite capacity networks, which are encountered in practice.

## 4.2 Limitations

Similarly to Kadkhodaie et al. (2024), our experiments, while thorough, focus on relatively small scale settings, which may not reflect the current state-of-the-art. This choice was deliberate: a rigorous validation of our claims at large scales is computationally prohibitive and would require significantly more resources and time. Ultimately, we have prioritized insight as opposed to completeness. Moreover, it is important to stress that we have specifically studied directional inductive biases imposed by the architecture. In principle, the overall dynamics of diffusion models may also be influenced or dominated by different factors such as explicit regularization or other implicit priors, e.g., the double descent phenomenon (Nakkiran et al., 2020).

## 4.3 Broader Impact

We investigate the directional inductive biases inherent in score-based generative models and examine their influence on the approximation capabilities of neural networks. We propose a method to systematically characterize these biases, offering insights into both the successes and potential limitations of modern diffusion models. Although our focus is primarily theoretical, understanding these mechanisms may guide the design of more effective and efficient generative modeling algorithms.

Specifically, deep learning is, at present, an empirical science that is enabled by scale and heavy reliance on heuristics. We believe that our theoretical insights and further developments along this line of research could allow for more cost-effective and principled development of generative technologies. For example, we see potential applications of our work in AutoML (Bergstra et al., 2011) and neural architecture search (Zoph & Le, 2017).

Importantly, characterizing the inductive biases of generative models has implications for understanding and mitigating undesirable behaviors such as memorization of training data (Carlini et al., 2023; Somepalli et al., 2023; Wen et al., 2024; Gu et al., 2025) and hallucination in generated outputs (Aithal et al., 2024; Lu et al., 2025; Floros et al., 2025). By making explicit which patterns the model is predisposed to reproduce, our approach could help identify circumstances under which models are likely to overfit or generate spurious content. This, in turn, may inform the development of techniques aimed at reducing privacy risks, improving reliability and enhancing the factual correctness of generated outputs. In this sense, our work contributes not only to the theoretical understanding of generative modeling but also to the broader goal of creating safer and more trustworthy systems.

## Reproducibility Statement

Code to reproduce our experiments is in the supplementary material, with details in Appendix A.

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

Table 1: Hyperparameters of iDDPM and DiT architectures used in the paper. *We train DiT/4 for double the iterations such that the running time is roughly the same as DiT/2 and iDDPM networks.

| iDDPM | $\mathcal{N}(\mathbf{0}, D\boldsymbol{vv}^\top)$ | MNIST | CelebA-HQ | CIFAR-10 | DiT | $\mathcal{N}(\mathbf{0}, D\boldsymbol{vv}^\top)$ | Sphere |
|---|---|---|---|---|---|---|---|
| shape | $1 \times 16 \times 16$ | $1 \times 28 \times 28$ | $1 \times 56 \times 56$ | $3 \times 32 \times 32$ | shape | $1 \times 16 \times 16$ | $1 \times 16 \times 16$ |
| diffusion_steps | 1000 | 1000 | 1000 | 1000 | diffusion_steps | 1000 | 1000 |
| noise_schedule | linear | linear | linear | linear | noise_schedule | linear | linear |
| channels | 32 | 32 | 32 | 32 | patch_size | 2/4 | 2 |
| channel_mults | 1, 1 | 1, 1 | 1, 1, 1 | 1, 1, 1 | hidden_size | 48 | 48 |
| depth | 1 | 2 | 2 | 2 | depth | 8 | 8 |
| attn_resolutions | - | - | - | - | mlp_ratio | 2 | 2 |
| num_heads | 4 | 4 | 4 | 4 | num_heads | 4 | 4 |
| batch_size | 1000 | 500 | 500 | 500 | batch_size | 1000 | 1000 |
| iterations | 2k | 100k | 100k | 200k | iterations | 2/4k* | 10k |
| learning_rate | 1e-4 | 1e-3 | 1e-3 | 1e-3 | learning_rate | 1e-4 | 1e-4 |

## A  EXPERIMENTAL SETUP

All of our experiments were conducted on a Linux machine with 128GB of RAM and a NVIDIA RTX 4090 GPU. We train and evaluate diffusion models according to the DDPM framework (Ho et al., 2020), with our hyperparameters included in Table 1. To compute the sliced Wasserstein metrics, we use an overcomplete set of $L = 64D$ random directions. All experiments are repeated five times with different random seeds. Each dataset we consider consists of 10k samples. For sphere modeling, shown in Figure 1, the data is aligned with the first three and the last three SADs. When estimating the average geometry, $\mathbf{G}_\mathcal{F}(\mathcal{P}, \Theta)$, we use 1M randomly initialized networks.

## B  REDISCOVERING GEOMETRY-ADAPTIVE HARMONIC BASES

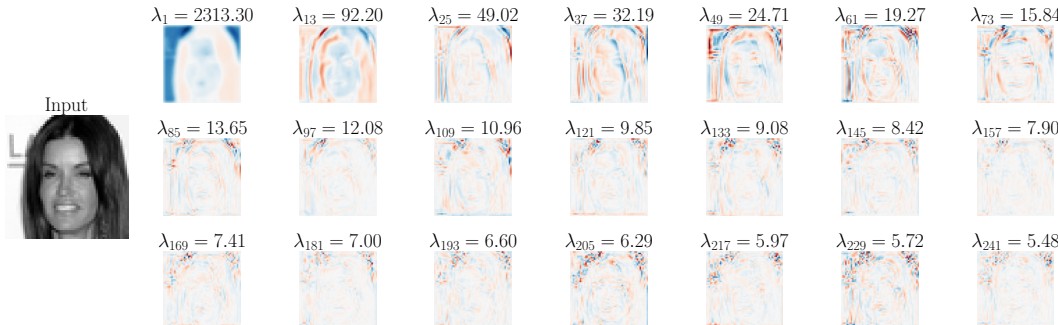

Figure 9: Eigendecomposition of iDDPM average geometry, $\mathbf{G}_\mathcal{F}(\mathcal{P}, \Theta)$, at initialization with probing distribution $\mathcal{P} = \mathcal{N}(\boldsymbol{x}, \sigma^2 \boldsymbol{I}) \times \mathcal{U}(\{\sigma_{\min}, \dots, \sigma_{\max}\})$, i.e., we probe along the standard forward diffusion process with the data sample $\boldsymbol{x}$. The input image, $\boldsymbol{x}$, is shown on the left and the first few eigenvectors, together with the corresponding unnormalized eigenvalues, are shown on the right. Despite our approach being fundamentally different from existing works, the results resemble the GAHBs of Kadkhodaie et al. (2024). Specifically, we see that our eigenvectors are also adaptive to the geometry of the input image. Moreover, they have harmonic structure as we also observe oscillating patterns whose frequency increases as the eigenvalue decreases.

## C  GEOMETRIES OF COMMON NEURAL NETWORK ARCHITECTURES

**Proposition 1** (MLP geometry, proof in Appendix D.4). *Let $\mathcal{F}$ be the family of networks of the form $\boldsymbol{z} = \phi(\boldsymbol{W}\boldsymbol{h} + \boldsymbol{b})$, where $\phi : \mathbb{R} \to \mathbb{R}$ is applied element-wise and $\boldsymbol{h} \in \mathbb{R}^L$ is an arbitrary function of the input. Assume that $\boldsymbol{W}, \boldsymbol{b}$ are completely independent and parameters within each group are identically distributed. Letting $\mathbf{1}$ represent a vector of ones, the geometry takes the form:*

$$\mathbf{G}_\mathcal{F}(\mathcal{P}, \Theta) = \alpha_\mathcal{F}(\mathcal{P}, \Theta)\boldsymbol{I} + \beta_\mathcal{F}(\mathcal{P}, \Theta)\mathbf{1}\mathbf{1}^\top. \tag{7}$$

**Proposition 2** (CNN geometry, proof in Appendix D.5). *Let $\mathcal{F}$ be the family of networks of the form $\boldsymbol{z} = \phi(\boldsymbol{W}\boldsymbol{h} + \boldsymbol{b})$, where $\phi : \mathbb{R} \to \mathbb{R}$ is applied element-wise and $\boldsymbol{h} \in \mathbb{R}^{C_{in} \times L}$ denotes an arbitrary function of the input. $\boldsymbol{W}$ represents convolution with kernel size $k$, $C_{in}$ input channels and $C_{out}$ output channels. Assume $\boldsymbol{W}, \boldsymbol{b}$ are completely independent and parameters within each group are identically distributed. Then, letting $\otimes$ denote the Kronecker product, the geometry is a block-diagonal matrix with a block-constant perturbation:*

$$\mathbf{G}_{\mathcal{F}}(\mathcal{P}, \Theta) = \boldsymbol{I}_{C_{out}} \otimes \boldsymbol{A}_{\mathcal{F}}(\mathcal{P}, \Theta) + (\mathbf{1}\mathbf{1}^{\top})_{C_{out}} \otimes \boldsymbol{B}_{\mathcal{F}}(\mathcal{P}, \Theta). \tag{8}$$

**Proposition 3** (Transformer geometry, proof in Appendix D.6). *Let $\mathcal{F}$ be the family of networks of the form $\boldsymbol{z} = \boldsymbol{Q}(\boldsymbol{W}\boldsymbol{h} + \boldsymbol{b})$, where $\boldsymbol{Q}$ is fixed, orthonormal (e.g., unpatchify) and $\boldsymbol{h} \in \mathbb{R}^{T \times L_{in}}$ denotes an arbitrary function of the input. $\boldsymbol{W} : \mathbb{R}^{T \times L_{in}} \to \mathbb{R}^{T \times L_{out}}$ and $\boldsymbol{b}$ represent a transformer-style layer operating on $T$ tokens separately. Assume $\boldsymbol{W}, \boldsymbol{b}$ are completely independent and parameters within each group are zero-mean, identically distributed. Then, the geometry has at most $T$ distinct eigenvalues and takes the form:*

$$\mathbf{G}_{\mathcal{F}}(\mathcal{P}, \Theta) = \boldsymbol{Q}[\boldsymbol{A}_{\mathcal{F}}(\mathcal{P}, \Theta) \otimes \boldsymbol{I}_{L_{out}}]\boldsymbol{Q}^{\top}. \tag{9}$$

# D  DEFERRED PROOFS

## D.1  LEMMAS

**Lemma 1.** *Consider DSM with data drawn from $\mathcal{N}(\mathbf{0}, \boldsymbol{v}\boldsymbol{v}^{\top})$ for a fixed noise level $\sigma > 0$ and $\|\boldsymbol{v}\|_2 = 1$. The associated score function is linear and of the form $\boldsymbol{\Omega}(\cdot)$, with $\boldsymbol{\Omega} = \frac{1}{\sigma^2}\left(\frac{\boldsymbol{v}\boldsymbol{v}^{\top}}{\sigma^2+1} - \boldsymbol{I}\right)$.*

*Proof.* We have the density $p_{\sigma} = \mathcal{N}(\mathbf{0}, \boldsymbol{v}\boldsymbol{v}^{\top} + \sigma^2\boldsymbol{I})$. The log-density is therefore quadratic, resulting in a linear score function as follows:

$$\nabla_{\boldsymbol{x}_{\sigma}} \log p_{\sigma}(\boldsymbol{x}_{\sigma}) = -\frac{1}{2}\nabla_{\boldsymbol{x}_{\sigma}}\left[\boldsymbol{x}_{\sigma}^{\top}(\boldsymbol{v}\boldsymbol{v}^{\top} + \sigma^2\boldsymbol{I})^{-1}\boldsymbol{x}_{\sigma}\right] = -(\boldsymbol{v}\boldsymbol{v}^{\top} + \sigma^2\boldsymbol{I})^{-1}\boldsymbol{x}_{\sigma}. \tag{10}$$

We can further simplify this via the Sherman-Morrison formula:

$$\boldsymbol{\Omega} = -(\boldsymbol{v}\boldsymbol{v}^{\top} + \sigma^2\boldsymbol{I})^{-1} = \frac{1}{\sigma^2}\left(\frac{\boldsymbol{v}\boldsymbol{v}^{\top}}{\sigma^2+1} - \boldsymbol{I}\right). \tag{11}$$

$\square$

**Lemma 2.** *Let $\boldsymbol{W}$ be a random matrix with entries that are iid, with mean zero and variance $\sigma^2$. For any compatible matrix $\boldsymbol{X}$, $\mathbb{E}_{\boldsymbol{W}}[\boldsymbol{W}^{\top}\boldsymbol{X}\boldsymbol{W}] = \sigma^2 \operatorname{tr}(\boldsymbol{X})\boldsymbol{I}$, where $\operatorname{tr}(\cdot)$ is the trace.*

*Proof.* Let $(\cdot)^{(i)}$ be column $i$. For entry $(i, j)$ we can write the expectation as follows:

$$\sum_{k} \mathbb{E}\left[\boldsymbol{W}^{(i)\top}\boldsymbol{X}^{(k)}\boldsymbol{W}_{(k,j)}\right] = \sum_{k} \sigma^2 \delta_{i-j}\boldsymbol{X}_{(k,k)} = \sigma^2 \operatorname{tr}(\boldsymbol{X})\delta_{i-j}. \tag{12}$$

$\square$

## D.2 THEOREM 1

*Proof.* Let $\boldsymbol{\Omega} = \boldsymbol{\Phi}\boldsymbol{\Theta}$ be a linear model. The optimization objective is:

$$
\begin{aligned}
\mathcal{J}_{\text{DSM}}(\boldsymbol{\Theta}) &= \mathbb{E}_{\boldsymbol{x}\sim\mathcal{N}(\boldsymbol{0},\boldsymbol{v}\boldsymbol{v}^\top),\boldsymbol{\epsilon}\sim\mathcal{N}(\boldsymbol{0},\boldsymbol{I})} \left[ \left\| \boldsymbol{\Phi}\boldsymbol{\Theta}(\boldsymbol{x}+\sigma\boldsymbol{\epsilon}) + \frac{\boldsymbol{\epsilon}}{\sigma} \right\|_2^2 \right] \\
&= \mathbb{E}\left[ \left\| \boldsymbol{\Phi}\boldsymbol{\Theta}\boldsymbol{x} + \left(\sigma\boldsymbol{\Phi}\boldsymbol{\Theta} + \frac{\boldsymbol{I}}{\sigma}\right)\boldsymbol{\epsilon} \right\|_2^2 \right] \\
&\overset{(*)}{=} \mathbb{E}\left[ \|\boldsymbol{\Phi}\boldsymbol{\Theta}\boldsymbol{x}\|_2^2 \right] + \mathbb{E}\left[ \left\| \left(\sigma\boldsymbol{\Phi}\boldsymbol{\Theta} + \frac{\boldsymbol{I}}{\sigma}\right)\boldsymbol{\epsilon} \right\|_2^2 \right] = \|\boldsymbol{\Phi}\boldsymbol{\Theta}\boldsymbol{v}\|_2^2 + \left\| \sigma\boldsymbol{\Phi}\boldsymbol{\Theta} + \frac{\boldsymbol{I}}{\sigma} \right\|_F^2,
\end{aligned}
\tag{13}
$$

where $(*)$ follows by independence of $\boldsymbol{x}$ and $\boldsymbol{\epsilon}$. The gradient with respect to $\boldsymbol{\Theta}$ is then given by:

$$
\nabla_{\boldsymbol{\Theta}} \mathcal{J}_{\text{DSM}}(\boldsymbol{\Theta}) = 2\boldsymbol{\Phi}^\top[\boldsymbol{\Phi}\boldsymbol{\Theta}(\boldsymbol{v}\boldsymbol{v}^\top + \sigma^2\boldsymbol{I}) + \boldsymbol{I}].
\tag{14}
$$

With this, and for a suitable $\eta > 0$, we express the GD learning dynamics with respect to $\boldsymbol{\Omega}$ as:

$$
\boldsymbol{\Omega}_t = \boldsymbol{\Omega}_{t-1} - \eta\boldsymbol{\Phi}\nabla_{\boldsymbol{\Theta}}\mathcal{J}_{\text{DSM}}(\boldsymbol{\Theta}) = \boldsymbol{\Omega}_{t-1} - 2\eta\boldsymbol{\Phi}\boldsymbol{\Phi}^\top[\boldsymbol{\Omega}_{t-1}(\boldsymbol{v}\boldsymbol{v}^\top + \sigma^2\boldsymbol{I}) + \boldsymbol{I}].
\tag{15}
$$

Moreover, we study the error dynamics defined by $\boldsymbol{E}_t = \boldsymbol{\Omega}_t - \boldsymbol{\Omega}^*$. Here, $\boldsymbol{\Omega}^* = \frac{1}{\sigma^2}\left(\frac{\boldsymbol{v}\boldsymbol{v}^\top}{\sigma^2+1} - \boldsymbol{I}\right)$ is given by Lemma 1 and we can verify that $\boldsymbol{\Omega}^*(\boldsymbol{v}\boldsymbol{v}^\top + \sigma^2\boldsymbol{I}) + \boldsymbol{I} = \boldsymbol{0}$, so it is a stationary point. Therefore, we write:

$$
\boldsymbol{E}_t = \boldsymbol{E}_{t-1} - 2\eta\boldsymbol{\Phi}\boldsymbol{\Phi}^\top\boldsymbol{E}_{t-1}(\boldsymbol{v}\boldsymbol{v}^\top + \sigma^2\boldsymbol{I}).
\tag{16}
$$

Let us focus on the sequence of expected errors, $\mathbb{E}[\boldsymbol{E}_t]$, where the randomness is over the initialization (and potentially stochastic gradients). Assuming $\mathbb{E}[\boldsymbol{\Theta}_0] = \boldsymbol{0}$, $\mathbb{E}[\boldsymbol{E}_0] = -\boldsymbol{\Omega}^*$. Additionally, if $\boldsymbol{v} \in \{\boldsymbol{u}_i\}_{i=1}^D$ is an eigenvector of $\boldsymbol{\Phi}\boldsymbol{\Phi}^\top$ with corresponding eigenvalues $\{\lambda_i\}_{i=1}^D$, matrices in Equation 16 commute since they share eigenvectors. This also forces the eigenspaces of all subsequent error terms. In particular, we have:

$$
\mathbb{E}[\boldsymbol{E}_t] = \mathbb{E}[\boldsymbol{E}_{t-1}] - 2\eta\boldsymbol{\Phi}\boldsymbol{\Phi}^\top(\boldsymbol{u}_i\boldsymbol{u}_i^\top + \sigma^2\boldsymbol{I})\mathbb{E}[\boldsymbol{E}_{t-1}] = -[\boldsymbol{I} - 2\eta(\lambda_i\boldsymbol{u}_i\boldsymbol{u}_i^\top + \sigma^2\boldsymbol{\Phi}\boldsymbol{\Phi}^\top)]^t\boldsymbol{\Omega}^*,
\tag{17}
$$

where it is clear that the error decays exponentially. For sufficiently small $\eta$, the iterated matrix is positive semidefinite and therefore convergence depends on the minimum eigenvalue of $\lambda_i\boldsymbol{u}_i\boldsymbol{u}_i^\top + \sigma^2\boldsymbol{\Phi}\boldsymbol{\Phi}^\top$, i.e., it is $\mathcal{O}[(1 - 2\eta\rho_i)^t]$ with $\rho_i = \min[(\sigma^2+1)\lambda_i, \sigma^2\min_{j\neq i}\lambda_j]$. Suppose $\exists\lambda_j < \lambda_i$, then $\rho_i = \sigma^2\lambda_D$, i.e., the convergence rate is fixed for $i < D$. However, for $i = D$ we have $\rho_D = \min[(\sigma^2+1)\lambda_D, \sigma^2\lambda_{D-1}] > \sigma^2\lambda_D$. That is, we converge faster for $i = D$.

Now, to complete the proof, we focus on the stochastic gradient at optimality:

$$
\nabla_{\boldsymbol{\Theta}}\widehat{\mathcal{J}}_{\text{DSM}}(\boldsymbol{x}, \boldsymbol{\epsilon}; \boldsymbol{\Theta}^*) = 2\boldsymbol{p}\boldsymbol{q}^\top, \quad \boldsymbol{p} = \boldsymbol{\Phi}^\top\left(\boldsymbol{\Omega}^*\boldsymbol{q} + \frac{\boldsymbol{\epsilon}}{\sigma}\right), \quad \boldsymbol{q} = \boldsymbol{x} + \sigma\boldsymbol{\epsilon}.
\tag{18}
$$

Note, by construction, all of the above quantities are zero-mean. In particular, this implies that $\boldsymbol{p}, \boldsymbol{q}$ are uncorrelated and, since they are jointly Gaussian, we claim that they are independent. Therefore, by setting $\boldsymbol{v} = \boldsymbol{u}_i$ and vectorizing, we can write the stochastic gradient covariance as:

$$
\begin{aligned}
4\mathbb{E}\left[\text{vec}(\boldsymbol{p}\boldsymbol{q}^\top)\text{vec}(\boldsymbol{p}\boldsymbol{q}^\top)^\top\right] &= 4\mathbb{E}\left[(\boldsymbol{q}\boldsymbol{q}^\top)\otimes(\boldsymbol{p}\boldsymbol{p}^\top)\right] \\
&= 4\mathbb{E}\left[\boldsymbol{q}\boldsymbol{q}^\top\right]\otimes\mathbb{E}\left[\boldsymbol{p}\boldsymbol{p}^\top\right] \\
&= \frac{4}{\sigma^2(\sigma^2+1)}(\boldsymbol{u}_i\boldsymbol{u}_i^\top + \sigma^2\boldsymbol{I})\otimes(\boldsymbol{\Phi}^\top\boldsymbol{u}_i\boldsymbol{u}_i^\top\boldsymbol{\Phi}) \propto \lambda_i.
\end{aligned}
\tag{19}
$$

$\square$

### D.3 THEOREM 2

*Proof.* We first simplify Equation 6:

$$\alpha = \mathbb{E}_{\boldsymbol{x} \sim p}[\boldsymbol{x}^\top(\boldsymbol{W}^\top \mathbf{G}_{\mathcal{F}} \boldsymbol{W} \boldsymbol{x})] = \mathrm{tr}(\boldsymbol{W}^\top \mathbf{G}_{\mathcal{F}} \boldsymbol{W} \boldsymbol{C}), \tag{20}$$

where $\boldsymbol{C} := \mathbb{E}_{\boldsymbol{x} \sim p}[\boldsymbol{x}\boldsymbol{x}^\top]$ is the second moment of the data. Let $\mathbf{G}_{\mathcal{F}} = \boldsymbol{U}\boldsymbol{\Lambda}\boldsymbol{U}^\top$ and $\boldsymbol{C} = \boldsymbol{V}\boldsymbol{\Sigma}\boldsymbol{V}^\top$ be eigendecompositions with eigenvalues $\{\lambda_i\}_{i=1}^D$ and $\{\sigma_i\}_{i=1}^D$ respectively. By defining the orthogonal matrix $\boldsymbol{Q} = \boldsymbol{U}^\top \boldsymbol{W} \boldsymbol{V}$, Equation 20 can be equivalently stated as follows:

$$\begin{aligned}
\alpha &= \mathrm{tr}(\boldsymbol{W}^\top \boldsymbol{U}\boldsymbol{\Lambda}\boldsymbol{U}^\top \boldsymbol{W} \boldsymbol{V}\boldsymbol{\Sigma}\boldsymbol{V}^\top) \\
&= \mathrm{tr}(\boldsymbol{V}^\top \boldsymbol{W}^\top \boldsymbol{U}\boldsymbol{\Lambda}\boldsymbol{U}^\top \boldsymbol{W} \boldsymbol{V}\boldsymbol{\Sigma}) \\
&= \mathrm{tr}(\boldsymbol{Q}^\top \boldsymbol{\Lambda} \boldsymbol{Q} \boldsymbol{\Sigma}) = \sum_{i=1}^D \sum_{j=1}^D \lambda_i \sigma_j [\boldsymbol{Q}_{(i,j)}]^2.
\end{aligned} \tag{21}$$

Observe that, since $\boldsymbol{Q}$ is orthogonal, the matrix defined by $\boldsymbol{P}_{(i,j)} = [\boldsymbol{Q}_{(i,j)}]^2$ is doubly stochastic. Moreover, Equation 21 shows that optimizing $\alpha$ is a linear problem in $\boldsymbol{P}$ over the the convex set of doubly stochastic matrices. In particular, by Birkhoff's theorem, $\alpha$ is extremized when $\boldsymbol{P}$ is a permutation matrix. The minimizing permutation is the one that misaligns the eigenvalues, i.e., $\boldsymbol{P} = \boldsymbol{R}$ is achieved if $\boldsymbol{Q} = \boldsymbol{R} \iff \boldsymbol{W} = \boldsymbol{U}\boldsymbol{R}\boldsymbol{V}^\top$. Similarly, $\alpha$ is maximized when the eigenvalues are aligned, i.e., $\boldsymbol{P} = \boldsymbol{I}$ if $\boldsymbol{Q} = \boldsymbol{I} \iff \boldsymbol{W} = \boldsymbol{U}\boldsymbol{V}^\top$. Indeed, one verifies that these permutations are optimal as they satisfy von Neumann's trace inequalities with equality. $\square$

### D.4 PROPOSITION 1

*Proof.* Let $\Theta$ denote the specified parameter distribution and consider a probe $\mathcal{P}$. The geometry is:

$$\mathbf{G}_{\mathcal{F}}(\mathcal{P}, \Theta) = \mathbb{E}_{\boldsymbol{h}}\mathbb{E}_{\boldsymbol{W}, \boldsymbol{b}}\left[\phi(\boldsymbol{W}\boldsymbol{h} + \boldsymbol{b})\phi(\boldsymbol{W}\boldsymbol{h} + \boldsymbol{b})^\top\right]. \tag{22}$$

First, compute the inner expectation, i.e., $\mathbf{G}_{\mathcal{F}}(\mathcal{P}, \Theta)|_{\boldsymbol{h}}$. For indices $i, j$, since the parameters are iid:

$$[\mathbf{G}_{\mathcal{F}}(\mathcal{P}, \Theta)|_{\boldsymbol{h}}]_{(i,j)} = \begin{cases} \mathbb{E}_{\boldsymbol{W}, \boldsymbol{b}}[\phi(\boldsymbol{W}^{(1)}\boldsymbol{h} + \boldsymbol{b}^{(1)})^2] & \text{if } i = j \\ \mathbb{E}_{\boldsymbol{W}, \boldsymbol{b}}[\phi(\boldsymbol{W}^{(1)}\boldsymbol{h} + \boldsymbol{b}^{(1)})]^2 & \text{otherwise} \end{cases}. \tag{23}$$

Writing $\boldsymbol{z}^{(1)} = \phi(\boldsymbol{W}^{(1)}\boldsymbol{h} + \boldsymbol{b}^{(1)})$, we can express the above via the conditional mean, $\mu_{\boldsymbol{z}^{(1)}|\boldsymbol{h}}$, and conditional variance $\sigma^2_{\boldsymbol{z}^{(1)}|\boldsymbol{h}}$:

$$\mathbf{G}_{\mathcal{F}}(\mathcal{P}, \Theta)|_{\boldsymbol{h}} = \sigma^2_{\boldsymbol{z}^{(1)}|\boldsymbol{h}}\boldsymbol{I} + \mu^2_{\boldsymbol{z}^{(1)}|\boldsymbol{h}}\mathbf{1}\mathbf{1}^\top, \tag{24}$$

where $\mathbf{1}$ is a vector of ones. Now, taking the outer expectation yields the desired result:

$$\mathbf{G}_{\mathcal{F}}(\mathcal{P}, \Theta) = \mathbb{E}_{\boldsymbol{h}}[\sigma^2_{\boldsymbol{z}^{(1)}|\boldsymbol{h}}]\boldsymbol{I} + \mathbb{E}_{\boldsymbol{h}}[\mu^2_{\boldsymbol{z}^{(1)}|\boldsymbol{h}}]\mathbf{1}\mathbf{1}^\top. \tag{25}$$

$\square$

### D.5 PROPOSITION 2

*Proof.* Similarly to our treatment of MLPs in Appendix D.4, we first compute the geometry conditioned on $\boldsymbol{h}$, i.e., $\mathbf{G}(\mathcal{P}, \Theta)|_{\boldsymbol{h}}$. Vectorizing, write the $i^{\text{th}}$ input channel as $\boldsymbol{h}^{(i)} \in \mathbb{R}^L$. Then, $\boldsymbol{W}$ is a $C_{\text{out}} \times C_{\text{in}}$ block matrix with each $\boldsymbol{W}_{(i,j)}$ representing convolution with a filter of size $k$. Therefore, $\boldsymbol{z}^{(i)} = \phi(\sum_j \boldsymbol{W}_{(i,j)}\boldsymbol{h}^{(j)} + \boldsymbol{b}^{(i)})$. Since the parameters are iid, the conditional geometry is expressed as the following $C_{\text{out}} \times C_{\text{out}}$ block matrix:

$$[\mathbf{G}_{\mathcal{F}}(\mathcal{P},\Theta)|_{\boldsymbol{h}}]_{(m,n)} = \begin{cases} \mathbb{E}_{\boldsymbol{W},\boldsymbol{b}}[\boldsymbol{z}^{(1)}\boldsymbol{z}^{(1)^{\top}}] & \text{if } m=n \\ \mathbb{E}_{\boldsymbol{W},\boldsymbol{b}}[\boldsymbol{z}^{(1)}]\mathbb{E}_{\boldsymbol{W},\boldsymbol{b}}[\boldsymbol{z}^{(1)}]^{\top} & \text{otherwise} \end{cases}. \tag{26}$$

We can rephrase this result in terms of the conditional mean, $\boldsymbol{\mu}_{\boldsymbol{z}^{(1)}|\boldsymbol{h}}$, and the conditional covariance $\boldsymbol{\Sigma}_{\boldsymbol{z}^{(1)}|\boldsymbol{h}}$. The manipulation is identical to the one used to derive the MLP geometry. Averaging $\boldsymbol{h}$:

$$\mathbf{G}_{\mathcal{F}}(\mathcal{P},\Theta) = \boldsymbol{I}_{C_{\text{out}}} \otimes \mathbb{E}_{\boldsymbol{h}}[\boldsymbol{\Sigma}_{\boldsymbol{z}^{(1)}|\boldsymbol{h}}] + (\boldsymbol{1}\boldsymbol{1}^{\top})_{C_{\text{out}}} \otimes \mathbb{E}_{\boldsymbol{h}}[\boldsymbol{\mu}_{\boldsymbol{z}^{(1)}|\boldsymbol{h}}\boldsymbol{\mu}_{\boldsymbol{z}^{(1)}|\boldsymbol{h}}^{\top}]. \tag{27}$$

$\square$

## D.6 PROPOSITION 3

*Proof.* We first focus on $\boldsymbol{Q} = \boldsymbol{I}$. After vectorizing $\boldsymbol{h}$, $\boldsymbol{W}$ is block-diagonal with $\boldsymbol{W}_{(i,i)} = \boldsymbol{W}_{(1,1)}$ operating on each token, which we write as $\boldsymbol{h}^{(i)} \in \mathbb{R}^{L_{\text{in}}}$. Conditioned on $\boldsymbol{h}$, the geometry is a block matrix with block $(i,j)$:

$$\begin{aligned} [\mathbf{G}_{\mathcal{F}}(\mathcal{P},\Theta)|_{\boldsymbol{h}}]_{(i,j)} &= \mathbb{E}_{\boldsymbol{W},\boldsymbol{b}}[(\boldsymbol{W}_{(1,1)}\boldsymbol{h}^{(i)} + \boldsymbol{b}^{(1)})(\boldsymbol{W}_{(1,1)}\boldsymbol{h}^{(j)} + \boldsymbol{b}^{(1)})^{\top}] \\ &= (\sigma_{\boldsymbol{W}}^2 \boldsymbol{h}^{(i)^{\top}}\boldsymbol{h}^{(j)} + \sigma_{\boldsymbol{b}}^2)\boldsymbol{I}, \end{aligned} \tag{28}$$

where the last equality is by Lemma 2, assuming parameters in $\boldsymbol{W}$, $\boldsymbol{b}$ have variances $\sigma_{\boldsymbol{W}}^2$, $\sigma_{\boldsymbol{b}}^2$ respectively. Notice that every block is $\propto \boldsymbol{I}$ and depends on entries of $\boldsymbol{h}\boldsymbol{h}^{\top} \in \mathbb{R}^{T \times T}$, where, with a slight abuse of notation we have reverted to the matrix representation $\boldsymbol{h} \in \mathbb{R}^{T \times L_{in}}$. Writing this compactly as $(\sigma_{\boldsymbol{W}}^2 \boldsymbol{h}\boldsymbol{h}^{\top} + \sigma_{\boldsymbol{b}}^2\boldsymbol{1}\boldsymbol{1}^{\top}) \otimes \boldsymbol{I}_{L_{\text{out}}}$, applying $\boldsymbol{Q}$ and averaging over $\boldsymbol{h}$ yields:

$$\mathbf{G}_{\mathcal{F}}(\mathcal{P},\Theta) = \boldsymbol{Q}\mathbb{E}_{\boldsymbol{h}}[(\sigma_{\boldsymbol{W}}^2 \boldsymbol{h}\boldsymbol{h}^{\top} + \sigma_{\boldsymbol{b}}^2\boldsymbol{1}\boldsymbol{1}^{\top}) \otimes \boldsymbol{I}_{L_{\text{out}}}]\boldsymbol{Q}^{\top}. \tag{29}$$

The eigenvalues of the above are products of the eigenvalues of the factors in the Kronecker product, i.e., they are the $T$ eigenvalues of $\sigma_{\boldsymbol{W}}^2\mathbb{E}_{\boldsymbol{h}}[\boldsymbol{h}\boldsymbol{h}^{\top}] + \sigma_{\boldsymbol{b}}^2\boldsymbol{1}\boldsymbol{1}^{\top}$.

$\square$

# E  MORE VISUALIZATIONS

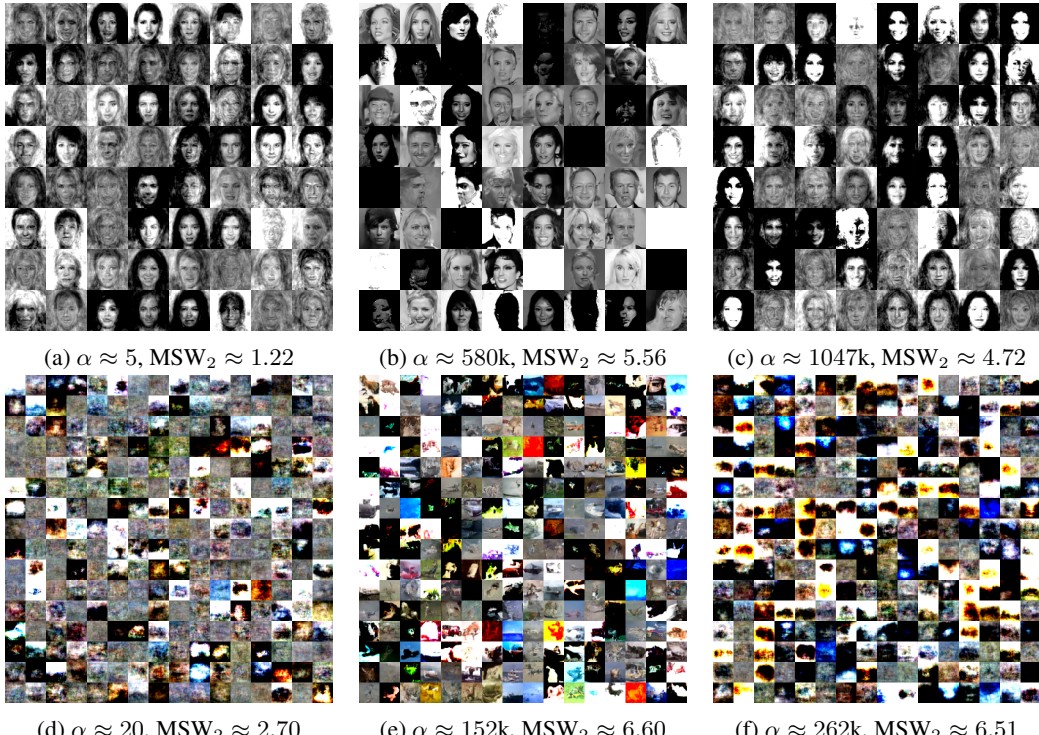

(a) $\alpha \approx 5$, MSW$_2 \approx 1.22$  (b) $\alpha \approx 580$k, MSW$_2 \approx 5.56$  (c) $\alpha \approx 1047$k, MSW$_2 \approx 4.72$

(d) $\alpha \approx 20$, MSW$_2 \approx 2.70$  (e) $\alpha \approx 152$k, MSW$_2 \approx 6.60$  (f) $\alpha \approx 262$k, MSW$_2 \approx 6.51$

Figure 10: Uncurated samples of iDDPMs trained on CelebA-HQ (top) and CIFAR-10 (bottom). We vary the alignment of the data with the geometry, $\alpha := \mathbb{E}_{\boldsymbol{x} \sim p}[\boldsymbol{z}^\top \mathbf{G}_{\mathcal{F}} \boldsymbol{z}]$, by transforming the data via appropriate matrices $\boldsymbol{W}$. The left column corresponds to minimal $\alpha$, the middle to the default and the right to maximum $\alpha$. Similarly with the samples of Figure 7, we see that the default alignment may lead to blank / saturated images. However, beyond this observation, it is difficult to draw conclusions based on perceptual quality since it is, by definition, subjective and, formally, does not imply an appropriate metric on distributions. A more meaningful, and statistically sound, comparison is via the Wasserstein metrics which validate our central hypothesis that low alignment leads to the best performance. Note that the networks are thoroughly trained. However, as our focus is on theoretical insight as opposed to completeness, there are several tricks (e.g., larger networks, more data, EMA, clipping) we have not employed that could lead to better quality results.

# F  SENSITIVITY OF $\mathbf{G}_{\mathcal{F}}$

In Section 3.2 we state that the probing distribution, $\mathcal{P}$, for the average geometry, $\mathbf{G}_{\mathcal{F}}(\mathcal{P}, \Theta)$, is a non-critical hyperparameter and not central to our analysis. Here we explore how injecting anisotropy via $\mathcal{P}$ might influence the underlying eigenvectors of $\mathbf{G}_{\mathcal{F}}(\mathcal{P}, \Theta)$ and our estimated SADs. To simplify matters and notation, we will assume that $\sigma$ is drawn uniformly from the noise levels of interest and the default initialization scheme of the architectures, as in Section 3.2. Under this setup, we will write the geometries for a given family of networks, $\mathcal{F}$, as $\mathbf{G}_{\mathcal{F}}(\mathcal{P})$ where $\mathcal{P}$ will capture some desired data probe.

For a fixed family (e.g., iDDPM or DiT) we consider degenerate, anisotropic probes $\mathcal{P}_i$. The data is drawn from $\delta_{\boldsymbol{z}_i}$ where $\boldsymbol{z}_i \sim \mathcal{N}(\mathbf{0}, \boldsymbol{I})$. That is, we first draw a random Gaussian vector and then probe with this fixed input to get a set of geometries $\{\mathbf{G}_{\mathcal{F}}(\mathcal{P}_i)\}_{i=1}^{N}$. Given this set, we would like to measure the average similarity of eigenvectors. One approach to this problem would be to consider the standard (squared) dot product between corresponding eigenvectors. That is, if $\{\boldsymbol{u}_d\}_{d=1}^{D}$ and

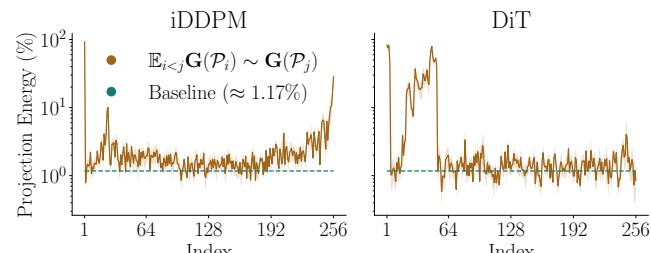

Figure 11: We investigate the robustness of the eigenvectors of $\mathbf{G}_{\mathcal{F}}(\mathcal{P}, \Theta)$ as a function of the probing distribution $\mathcal{P}$.

$\{\boldsymbol{v}_d\}_{d=1}^{D}$ are the eigenvectors of $\mathbf{G}_{\mathcal{F}}(\mathcal{P}_i)$ and $\mathbf{G}_{\mathcal{F}}(\mathcal{P}_j)$ respectively, one may compute $\{\boldsymbol{v}_d^{\top}\boldsymbol{u}_d\}_{d=1}^{D}$ and compare with a random baseline, e.g., the expected squared product between normalized Gaussian vectors is $1/D$. Note, however, that this method only gives a rough estimate of similarity as it does not account for eigen-multiplicity. However, it is straightforward to extend it to a window of size $2h + 1$. Concretely, denote $\boldsymbol{U} = [\boldsymbol{u}_1, \ldots, \boldsymbol{u}_D] \in \mathbb{R}^{D \times D}$. For a given vector, $\boldsymbol{v}_d$, we propose to compute the projection energies along the subspace defined by $2h + 1$ consecutive columns of $\boldsymbol{U}$. Representing columns starting from $a$ and ending at $b$ as $\boldsymbol{U}_{a:b} \in \mathbb{R}^{D \times b-a}$, we write:

$$\|\boldsymbol{U}_{a:b}\boldsymbol{U}_{a:b}^{\top}\boldsymbol{v}_d\|_2^2, \quad a, b, d \in \{1, \ldots, D\}, \quad b - a = 2h + 1, \quad a + b \approx 2d. \tag{30}$$

Note that these energies are $\in [0, 1]$ for orthonormal eigenbases and that high energy signals high similarity. Also note that this is a natural extension of cosine similarity, which is recovered by taking $h = 0$. In general, the random baseline for an arbitrary $h$ gives a similarity of $(2h + 1)/D$. Although this is an improved similarity measure compared to the standard dot product, we caution that it is imperfect and may still underestimate similarity for high eigen-multiplicity subspaces when $h$ is not properly callibrated. Nevertheless, our aim here is to show that despite such imperfections the underlying SADs (i.e., the eigenvectors of the geometry) are fairly robust even under extremely anisotropic probes.

In particular, we validate this claim in Figure 11 where we design our experiment for a space with $D = 256$ dimensions, pick $h = 1$ and quantify similarity over $N = 5$ different realizations of geometries. Interestingly, we find that, on average, the similarities are very high relative to the baseline (e.g. $\sim 100\%$ vs. $1.2\%$), especially for the first few (largest) eigenvectors. Even in the case of the last few (smallest) vectors, we see that the iDDPM U-Net still exhibits significant similarities. For DiT, as previously observed in Figure 6, the architecture shows considerable eigen-multiplicity for small eigenvalue vectors, i.e., $\{\boldsymbol{u}_d\}_{d=64}^{256}$ in Figure 11 are likely to belong to the same subspace. Therefore, although not explicitly shown in the figure due our choice of fixed, small $h$, given that that the first few vectors are similar, this forces high similarity over the rest of the spectrum as well.

