# OpenReview forum: "On the Anisotropy of Score-Based Generative Models"
_ICLR.cc/2026/Conference — Submitted to ICLR 2026_

### Official Review · Reviewer_UNQs · 2025-10-20

**Soundness:** 2
**Presentation:** 3
**Contribution:** 2
**Rating:** 4
**Confidence:** 3

**Summary:**

The authors study the inductive biases of neural networks trained on denoising score matching and hypothesize that these inductive biases can be captured by fixed linear bases. In an idealized setting, where the inner weights of a linear network $\Omega = \Phi \Theta$ are trained on denoising score matching for a rank one Gaussian $N(0,vv^T)$, the authors demonstrate that the inductive bias of the family of linear networks is described by the eigenvectors of $\Phi$. More generally, the authors define the average geometry of a family of score networks in terms of the averaged outer product, where the averaging is done over the parameter distribution and a ‘probing distribution’ on the inputs of the score network. The authors claim that the eigenvectors of this matrix capture the inductive biases of the family of score networks and provide experiments which evidence for this claim.

**Strengths:**

- Understanding the inductive biases of diffusion models is an integral problem in deep learning, and the authors provide some progress towards this endeavor.
- The ideas of the paper are generally architecture-independent.
- The paper is generally well-written and it is easy to understand their central claims.

**Weaknesses:**

- The novelty of the paper is limited, as the paper borrows significantly from previous attempts to formalize inductive biases in the discriminative setting [1], [2]. While the paper makes some improvements over previous work [3] in understanding inductive biases for diffusion models (e.g., moving beyond CNNs), some similar ideas are borrowed.
- I don’t find the central claim of the paper (i.e., that the average geometry describes the inductive bias of score networks) to be very well supported. Theoretically, it is essentially unprincipled except for some results in idealized settings, and the scope of the experiments is pretty narrow.

References:
[1] Ortiz-Jiménez, Guillermo, Apostolos Modas, Seyed-Mohsen Moosavi, and Pascal Frossard. "Neural anisotropy directions." Advances in Neural Information Processing Systems 33 (2020): 17896-17906.
[2] Radhakrishnan, Adityanarayanan, Daniel Beaglehole, Parthe Pandit, and Mikhail Belkin. "Mechanism for feature learning in neural networks and backpropagation-free machine learning models." Science 383, no. 6690 (2024): 1461-1467.
[3] Kadkhodaie, Zahra, Florentin Guth, Eero P. Simoncelli, and Stéphane Mallat. "Generalization in diffusion models arises from geometry-adaptive harmonic representations." arXiv preprint arXiv:2310.02557 (2023).

**Questions:**

I am confused by the statement of Theorem 1, specifically the line ‘the mean error to the optimal solution $v = u_i$ decays as…”. In what sense is $v = u_i$ an ‘optimal solution’, if the optimization is being done over $\Theta$? It seems like $v = u_i$ is an assumption of the theorem, and indeed the proof uses this fact in Equation (17). Also, I believe Equation (19) has an error, since the vectors $p$ and $q$ are not independent (although the proof claims that they are). In reality, I think you should have something like $$\mathbb{E}[\langle p, q \rangle] =  \mathbb{E}_{N(0,vv^T)}[x^T (\Omega^{\ast})^T \Phi x] + \sigma^2 \mathbb{E}[\epsilon^T (\Omega^{\ast})^T \Phi \epsilon] + \mathbb{E}[\epsilon^T \Phi \epsilon] > 0.$$

Thus, it is not clear to me whether the final sentence of Theorem 1 is true. Please let me know if I have misunderstood something here.

---

> ### Author Response · Authors · 2025-11-13
> **Thank you for your review (1/2)**
>
> Thank you for your review. We are pleased that you appreciated the importance of understanding inductive biases in diffusion and that you found our paper well-written and easy to understand.
>
> We address your concerns below.
>
> W1. Novelty
> -
>
> We respectfully disagree with your comment that "novelty of the paper is limited".
>
> **Please note that we have explicitly acknowledged and discussed the prior works you mentioned [1], [3] (Sections 2 and 4.1) and we have now cited [2] for completeness in Section 2**.
>
> Importantly, while it is true that our approach draws inspiration from [1], **to our knowledge there is no other work in the literature that has systematically described the directional inductive biases of diffusion models as we have. We believe, and, it seems, all other reviewers largely agree, that our contribution to the community is significant enough**.
>
> You then say that "similar ideas are borrowed" from [3]. Could you please clarify what you mean by this? **We strongly believe that our analysis is novel with respect to [3]**. [3] analyzes *trained*, homogeneous U-Nets and empirically shows that such networks are biased toward harmonic representations, only giving a loose characterization of inductive biases (as acknowledged by the authors themselves). In contrast, our analysis is:
>
> - architecture agnostic (as you also acknowledge in S2),
> - does not rely on Jacobians / homogeneneity,
> - provides a precise characterization of inductive biases by ordering the output space (via the SADs),
> - is consistent with and appears to recover results of [3] for the special case of convolutional networks (see Figure 9) and
> - empirically, provides a systematic way to predict generalization ability *prior* to training (i.e., our average geometry is predictive despite being calculated at initialization)
>
> Please note that the above points and comparisons are also included our paper, in Section 4.1.
>
> W2. Support for Conjecture 1
> -
>
> While it is true that we do not have a formal proof of our central conjecture (so we call it a conjecture), we do not agree that our approach is "unprincipled".
>
> Intuitively, we believe our Conjecture 1 holds because the score function models gradient steps. **A direction that is not easily estimated by the score function (i.e., not aligned with its geometry) is one where the reverse diffusion iterates do not change much so data along such directions persists**. In contrast, directions that are well modeled by the score (i.e., aligned with the geometry) will be significantly changed during reverse diffusion and so data along them will get "denoised" heavily. This intuition is mathematically precise and proven in the linear case (see Section 3.1 and Theorem 1). **The case of general networks is analyzed in Section 3.2**. In the latter case, **we also provide intuition and heuristics via a Markov bound (see Equation 4)** which leads to our central Conjecture 1 (experimentally validated in Figures 1, 6 and 8).
>
> Regarding your comment about our experiments being "pretty narrow":
>
> **We have validated our Conjecture 1 on synthetic data (spheres in Figure 1, rank-one data in Figure 6), common image benchmarks (MNIST, CIFAR-10, CelebA-HQ), across different architectures (iDDPM U-Nets with different resampling, DiTs with different patch sizes) and with five independent runs per experiment. In total, this amounts to training thousands of diffusion models** (e.g., Figure 6 shows four plots, each visualizing performance on 256 rank-one datasets, and we repeat for five runs for a total of 5120 trained models).
>
> In all these experiments, we only vary the alignment of the data with the architecture. In this sense, we have isolated the directional preferences as the cause of the observed optimization dynamics and final performances.
>
> For a number of reasons that we have listed in our rebuttal to Reviewer `rSjW` (see our answer to their W1), larger scale experiments are not feasible for us to perform; the issue is computational in nature.

---

> ### Author Response · Authors · 2025-11-13
> **Thank you for your review (2/2)**
>
> Q1. Theorem 1
> -
>
> The quote is not accurate, Theorem 1 reads "the mean error to the optimal solution *for* $\boldsymbol{v}=\boldsymbol{u}_i$ decays as...". So what this describes is the mean error to the optimal solution for the DSM problem with the data distribution $\mathcal{N}(\boldsymbol{0},\boldsymbol{u}_i\boldsymbol{u}^\top_i)$.
>
> Regarding your question about Equation (19) and the proof:
>
> We believe there is no mistake here. To show that $\boldsymbol{p}$ and $\boldsymbol{q}$ are independent we use the following facts:
>
> - $\boldsymbol{q}=\boldsymbol{x}+\sigma\boldsymbol{\epsilon}$. Therefore, we have $\mathbb{E}[\boldsymbol{q}]=\boldsymbol{0}$
> - $\boldsymbol{p}=\boldsymbol{\Phi}^\top(\boldsymbol{\Omega}^*\boldsymbol{q}+\boldsymbol{\epsilon}/\sigma)$. Therefore, we have $\mathbb{E}[\boldsymbol{p}]=\boldsymbol{0}$
> - The stochastic gradient at optimality is: $\nabla_{\boldsymbol{\Theta}}\hat{\mathcal{J}}=2\boldsymbol{p}\boldsymbol{q}^\top$. Since this is at optimality, we have $\mathbb{E}[\nabla_{\boldsymbol{\Theta}}\hat{\mathcal{J}}]=\boldsymbol{0}$. Therefore $\mathbb{E}[\boldsymbol{p}\boldsymbol{q}^\top]=\boldsymbol{0}=\mathbb{E}[\boldsymbol{p}]\mathbb{E}[\boldsymbol{q}]^\top$.
>
> What the above show is that $\boldsymbol{p}$ and $\boldsymbol{q}$ are uncorrelated. Moreover, since they are jointly Gaussian, as we have stated in our proof in Appendix D.2, they are indeed independent.
>
> ---
>
> **Once again, we would like to thank you for taking the time to review our paper. We hope that our answers and efforts are sufficient for you to consider raising your score and confidence in our work. We remain open to further discussion**.
>
> ---
>
> [1] *Neural Anisotropy Directions*, NeurIPS 2020
>
> [2] *Mechanism for feature learning in neural networks and backpropagation-free machine learning models*, Science 2024
>
> [3] *Generalization in diffusion models arises from geometry-adaptive harmonic representations*, ICLR 2024

---

### Official Review · Reviewer_NrCH · 2025-10-28

**Soundness:** 3
**Presentation:** 3
**Contribution:** 3
**Rating:** 4
**Confidence:** 2

**Summary:**

The paper introduces a new theoretical framework to analyze the directional inductive biases of score-based generative models (diffusion models). It defines Score Anisotropy Directions (SADs) — architecture-dependent directions in the model output space — which are said to describe how different neural network architectures (e.g., U-Nets, DiTs) preferentially model certain data structures. The authors propose the concept of an average geometry matrix whose eigendecomposition predicts which data directions are more easily learned. They provide both analytical derivations (for simplified settings) and empirical evidence on synthetic and image datasets to support the claim that alignment between data and architectural geometry determines generalization quality. The experiments mainly focus on Wasserstein distances as a measure of generative performance.

**Strengths:**

1. **Novel conceptual framing:**
The paper offers an original perspective by proposing SADs as a way to quantify architectural inductive biases in score-based generative models.
2. **Synthesis of architecture and anisotropy analysis:**
It connects the anisotropic conditioning of the optimization landscape to generative modeling performance, extending concepts like Neural Anisotropy Directions (NADs) from discriminative to generative contexts.
3. **Theoretical–empirical bridge:**
The authors combine analytical derivations for simplified cases with experimental validation on U-Net and DiT architectures, showing distinct anisotropy profiles across architectures.
4. **Insightful empirical angle:**
The work provides technical examples illustrating how geometric alignment affects generalization and how this connects to data augmentation effects in diffusion training.
5. **Potential interpretability contribution:**
The proposed framework could, in principle, help explain why certain architectures generalize better or exhibit specific generative artifacts.

**Weaknesses:**

1. **Conceptual overstatement and lack of rigor:**
Most results are heuristic and not rigorously proven. The central claims (Conjecture 1 and its supporting theorems) are largely empirical conjectures, not formal results. The link between anisotropy and generalization remains speculative.
2. **Theoretical fragility:**
The formal parts (e.g., Theorem 1) rely on linearized and oversimplified assumptions that are far removed from the nonlinear reality of diffusion networks. The claimed generalization to complex architectures is not mathematically justified.
3. **Limited experimental depth:**
The experiments use small-scale datasets (MNIST, CIFAR-10, CelebA-HQ) and low-dimensional toy problems. The evidence is insufficient to validate claims about “predicting generalization before training.”
4. **Ambiguous interpretation of metrics:**
The use of Wasserstein distances as the sole quantitative indicator of generative quality is weak — no perceptual or likelihood-based metrics are provided. The connection between these distances and “generalization” is asserted but not justified.
5. **Unclear practical relevance:**
While the average geometry can be computed at initialization, it is not demonstrated that this analysis provides actionable insights for model design or training.

**Questions:**

- The matrix $G_\mathcal{F} = \mathbb{E}[ F_\theta (x_\sigma, \sigma)F_\theta(x_\sigma, \sigma)^\top] $ depends on both $P$ and the initialization scheme. How sensitive are your results to these choices?
- In Theorem 1, the model is assumed linear. How do you justify extrapolating these linear results to complex, nonlinear diffusion architectures?
- The conjecture linking SADs to eigenvectors of  $G_\mathcal{F}$ is intuitive, but is there any formal result supporting it for nonlinear or trained networks?

---

> ### Author Response · Authors · 2025-11-13
> **Thank you for your review (1/2)**
>
> Thank you for your review. We are pleased that you found our analysis novel and insightful.
>
> We address your concerns below.
>
> W1/2. Conceptual overstatement & lack of rigor / Theoretical fragility
> -
>
> You claim that most of our results (supporting theorems) are heuristic and not rigorously proven. Could you please clarify what you mean by this? **All theorems, propositions and lemmas are formally proven in Appendix D. The only statement for which we do not provide a theoretical proof is Conjecture 1 (where we do *not* overclaim, hence our labeling)**.
>
> While it is true that our analysis in Theorem 1 does not reflect real world, highly non-linear settings, this analysis is useful to first build insight about simpler problems before tackling real world settings and formulating a conjecture. **Our analysis in Sections 3.2, 3.3 gives several theoretical and empirical insights in support of Conjecture 1. However, once again, we stress that we do *not* claim a formal proof of Conjecture 1**.
>
> W3. Limited experimental depth
> -
>
> While our claim of predicting generalization ability prior to training is indeed bold, we believe that it is defensible and our experiments convincingly support it:
>
> **We have validated our Conjecture 1 on synthetic data (spheres in Figure 1, rank-one data in Figure 6), common image benchmarks (MNIST, CIFAR-10, CelebA-HQ), across different architectures (iDDPM U-Nets with different resampling, DiTs with different patch sizes) and with five independent runs per experiment. In total, this amounts to training thousands of diffusion models** (e.g., Figure 6 shows four plots, each visualizing performance on 256 rank-one datasets, and we repeat for five runs for a total of 5120 trained models).
>
> In all these experiments, we only vary the alignment of the data with the architecture. In this sense, we have isolated the directional preferences as the cause of the observed optimization dynamics and final performances.
>
> For a number of reasons that we have listed in our rebuttal to Reviewer `rSjW` (see our answer to their W1), larger scale experiments are not feasible for us to perform; the issue is computational in nature. Still, **we believe our experiments are convincing and our claims are properly and explicitly qualified in the limitations Section 4.2**.
>
> W4. Ambiguous interpretation of metrics
> -
>
> Unlike Wasserstein distances, **perceptual "metrics" do *not* correspond to valid statistical distances and can be adversarially manipulated, i.e., good perceptual scores do *not* imply generalization [1, 2]**. In fact, **relying on them is likely to introduce confounders given that the underlying perceptual networks (e.g., Inception) will have their own inductive biases and preferred directions**.
>
> **By definition, perceptual scores are subjective. Our aim is to study optimization dynamics which are inherently objective**.
>
> Regarding you comment about the connection between Wasserstein distances and generalization: **If the Wasserstein metrics are zero this guarantees that the diffusion model has learned the desired distribution (since these are formal metrics). This is *not* the case for perceptual "metrics"**.
>
> W5. Unclear practical relevance
> -
>
> While we are not prepared to make claims about large scale SOTA or foundation models (as also mentioned in limitations Section 4.2), we believe our work offers several insights that could be practically relevant to explore further. For example, you have explicitly acknowledged that our work provides an "insightful empirical angle" (S4). Moreover, you also acknowledge the "potential interpretability contribution" (S5). We provide some further insights below (also see our broader impact Section 4.3):
>
> - Firstly, our work could inform the design of more efficient and cheap architectures. For example, we have already shown in Section 3.3 how a simple, architecture-aware transformation may drastically improve performance on standard image benchmarks. **It is therefore plausible that our insights could be used to build more effective autoencoders for latent diffusion by designing latent spaces that are tailored to the diffusion model's anisotropy**.
>
> - Another important angle is on **understanding and mitigating undesirable phenomena in diffusion models such as memorization of training data and hallucinations**. Research on understanding inductive biases of diffusion models, like ours, is directly relevant here as **we have presented a way to describe the patterns that diffusion models are more likely to learn / reproduce**.

---

> ### Author Response · Authors · 2025-11-13
> **Thank you for your review (2/2)**
>
> Q1. Sensitivity of $\mathbf{G}_{\mathcal{F}}$
> -
>
> **We have now performed additional experiments that show the robustness of the average geometry in Appendix F**.
>
> Interestingly, the geometry is fairly robust with respect to the probing distribution $\mathcal{P}$.
>
> For the initialization scheme, $\Theta$, there is recent work that studies a similar "geometry" in the context of discriminative models [3]. The authors of this work find that their geometry is fairly robust (they identify invariances and propose the so called *Geometric Invariance Hypothesis*). The fact that our SADs appear to be relevant even after thousands of training iterations (see Appendix A and training code for details) may suggest that a similar result holds for our notion of geometry.
>
> Q2. Justification for non-linear models
> -
>
> Intuitively, we believe our Conjecture 1 holds because the score function models gradient steps. **A direction that is not easily estimated by the score function (i.e., not aligned with its geometry) is one where the reverse diffusion iterates do not change much so data along such directions persists**. In contrast, directions that are well modeled by the score (i.e., aligned with the geometry) will be significantly changed during reverse diffusion and so data along them will get "denoised" heavily. This intuition is mathematically precise and proven in the linear case (see Section 3.1 and Theorem 1). **The case of general networks is analyzed in Section 3.2**. In the latter case, **we also provide intuition and heuristics via a Markov bound (see Equation 4)** which leads to our central Conjecture 1 (experimentally validated in Figures 1, 6 and 8).
>
> Q3. Formal proof of Conjecture 1
> -
>
> **We do not claim to have a formal proof of Conjecture 1 (so we call it a conjecture)**. However, we have provided some intuition and heuristics in our answer to your Q2 that we believe justify our claims.
>
> ---
>
> **Once again, we would like to thank you for taking the time to review our paper. We hope that our answers and efforts are sufficient for you to consider raising your score and confidence in our work. We remain open to further discussion**.
>
> ---
>
> [1] *A Note on the Inception Score*, ICMLW 2018
>
> [2] *Exposing flaws of generative model evaluation metrics and their unfair treatment of diffusion models*, NeurIPS 2023
>
> [3] *Geometric Inductive Biases of Deep Networks: The Role of Data and Architecture*, ICLR 2025

---

### Official Review · Reviewer_z2i3 · 2025-11-01

**Soundness:** 3
**Presentation:** 4
**Contribution:** 3
**Rating:** 6
**Confidence:** 2

**Summary:**

The paper studies directional inductive biases of score-based generative models. The authors introduce Score Anisotropy Directions (SADs), which is an set of orthonormal directions in the output space, obtained as the eigenvectors (smallest to largest) of an average geometry matrix. In a linear DSM setting, the authors show that SGD training leads to anisotropy and converging fastest along small eigenvalue directions. For the general nonlinear models, they conjecture the same ordering and support it with: (i) synthetic rank‑one datasets aligned to different bases; (ii) experiments where performance follows the eigenvector order for U‑Net/iDDPM and DiT. On real datasets, they vary an alignment measure between the data and the model's geometry via orthogonal transforms and find that lower alignment improves the performance, while high alignment underperforms or offers no gain over the baseline. The paper also analyzes average geometry matrix for common architectures and connects SADs to Geometry‑Adaptive Harmonic Bases.

**Strengths:**

Overall, I think this is a good empirical paper that proposes an interesting phenomenon for score-based generative models. I especially appreciate the clarity around SADs, how to quantify the "preferred directions" in generative models, and the demonstration that lower data–geometry alignment leads to better generation quality. The result is simple to grasp through the manifold hypothesis of data distribution, yet seems to broadly applicable across architectures and optimization algorithms.

**Weaknesses:**

The only weakness I can think of is that, while extensive experiments are conducted to verify the main conjecture, it is rigorously proved only for a linear DSM toy model. The paper would be stronger with an intuitive argument in a wide‑network/NTK or mean‑field limit showing why the SAD eigen‑ordering should persist and clarifying conditions under which their conjecture holds true.

**Questions:**

Could the authors provide intuition for why Conjecture 1 should hold for non‑linear networks, and clarify how your average geometry matrix relates to NTK in the wide‑network/linearization limit?

---

> ### Author Response · Authors · 2025-11-13
> **Thank you for your review**
>
> Thank you for your review. We are pleased that you found our presentation excellent and our overall analysis interesting.
>
> We address your concern below:
>
> W/Q1. Intuition for non-linear networks & connection with NTK/wide limit
> -
>
> Intuitively, we believe our Conjecture 1 holds because the score function models gradient steps. **A direction that is not easily estimated by the score function (i.e., not aligned with its geometry) is one where the reverse diffusion iterates do not change much so data along such directions persists**. In contrast, directions that are well modeled by the score (i.e., aligned with the geometry) will be significantly changed during reverse diffusion and so data along them will get "denoised" heavily. This intuition is mathematically precise and proven in the linear case (see Section 3.1 and Theorem 1). **The case of general networks is analyzed in Section 3.2**. In the latter case, **we also provide intuition and heuristics via a Markov bound (see Equation 4)** which leads to our central Conjecture 1 (experimentally validated in Figures 1, 6 and 8).
>
> You then ask regarding potential connections with NTK/wide limit:
>
> **To address your concern, we have added some discussion on NTK and future work in Section 4.1**.
>
> Since our work adapts the NAD framework from discriminative models [1], where some connections were later shown with NTK [2], it is plausible that part of this analysis also translates to our SADs. However, we do not (yet) have a mathematically precise argument to link our findings with NTK. We acknowledge that this is an interesting exploration for future work.
>
> We would also like to clarify the following: **In W1 you specifically asked for justification of why the eigen-ordering should persist in wide networks**. Please note that, **unlike NTK, our analysis does not assume infinite width and we have shown that our $\mathbf{G}_{\mathcal{F}}$ is a meaningful quantity even for relatively small networks** (see our Appendix A and code for specifics). In this sense, whether increasing network capacity changes the underlying SADs is is independent of our Conjecture 1. Still, **under standard assumptions of infinite width networks, i.e., where zero mean NNGPs emerge with some kernel $K(\boldsymbol{x},\sigma,\boldsymbol{x}',\sigma')$, the SADs and their ordering are stable as the average geometry converges to $\mathbb{E}_{(\boldsymbol{x},\sigma)\sim\mathcal{P}}[K(\boldsymbol{x},\sigma,\boldsymbol{x},\sigma)]$**.
>
> ---
>
> **Once again, we would like to thank you for taking the time to review our paper. We hope that our answers and efforts are sufficient for you to consider raising your score and confidence in our work. We remain open to further discussion**.
>
> ---
>
> [1] *Neural Anisotropy Directions*, NeurIPS 2020
>
> [2] *What can linearized neural networks actually say about generalization?*, NeurIPS 2021

---

### Official Review · Reviewer_hXcw · 2025-11-01

**Soundness:** 3
**Presentation:** 2
**Contribution:** 3
**Rating:** 4
**Confidence:** 3

**Summary:**

This paper proposes the framework of *Score Anisotropy Directions* (SADs) as a tool for a better understanding of how score-based generative models perform, by providing a rough prediction on how well the model will perform, according to the geometry of the model output and that of the given dataset. This framework identifies the intrinsic geometric bias which the model in use has, and tells us how well this aligns with the dataset geometry is an indicator of the performance.

**Strengths:**

This work presents a new lens for studying the performance of score-based generative models. It even offers some novel insights, such as a training recipe that applies appropriate rigid motions to align the data with the SADs, and the bold claim that subspaces with small eigenvalues are easier for the models to learn.
The central topic itself is a timely one, as the impact of generative models is rapidly growing, while we still have little understanding on how they work.
I can see this work serving as a great starting point for the geometric/inductive bias approach to the theoretical understanding of score-based generative models.

**Weaknesses:**

As much as I like the idea and the approach, I also have some questions and concerns about the current state of the paper.

The writing, especially in the introduction, has room for improvement. For example, while the introduction claims to provide a precise notion of "geometry" (around line numbers 46--48), the very first definition of SADs (Definition 1) is somewhat vague. How is a "preference" measured, and what does it mean to "generate data *along* a direction"?

I would also suggest moving the related works section to the front instead of keeping it at the end with the conclusions, as this would give readers stronger motivation to delve further into the paper.

Meanwhile, I find it quite unclear what the figures are meant to convey, even after reading the entire paper. I don't see what the 3D projection of the $\mathbb{R}^{256}$ space in Figure 1 is intended to depict, especially with the point clouds obscuring the geometry. It is also not clear how I should interpret the 2D plots in Figure 2 and 3. It would be helpful if the authors could further elaborate on these visualizations.

For further questions, please see below.

**Questions:**

1. In Theorem 1, is having $\lambda_{D-1} \neq \lambda_D$ an (implicit) assumption? Also, it would be helpful if the statements are written more precisely and rigorously, such as the "*mean error* to the optimal solution" and "SGD steps with respect to $\Theta$ *for $\boldsymbol{v} = \boldsymbol{u}_i$* have *covariance* $\propto \lambda_i$; what is the *mean error*, and how does a SGD step be "with respect to $\Theta$" and "for $\boldsymbol{v} = \boldsymbol{u}_i$" simultaneously, and what could it having a *covariance* mean? I personally think a theorem statement should be clearly understandable without referring to its proof.

2. I am not sure if Markov's inequality (4) is the right way to back up the claim of "small eigenvalue directions are easier to learn". After all, Markov's inequality is just an upper bound for the tail probability; it can say something about small eigenvalue directions as the right hand side will be small, but nothing can be inferred from it about the behavior along large eigenvalue directions.

3. The intuitive explanations in Section 3.2 seem reasonable, but only when considered locally around a point *$\boldsymbol{x}_\sigma$* that happens to have a large log-density. In defining the *average geometry*, we average out the "geometry" with respect to the distribution of *$\boldsymbol{x}_\sigma$,* which will smooth (or even worse, possibly cancel) out directions that are only meaningful in a local sense. Thus, although it seems like there is an explanation about this right after Definition 2, it is still unclear to me how the local intuition applies to the notion of average geometry. For example, in an extreme toy case where the data distribution is bimodal in $\mathbb{R}^2$, with each mode having its own "meaningful" directions that are orthogonal to those of the other mode, the resulting average geometry would be meaningless. Is there any intuition as to why we should expect the *average geometry* to possess a global semantic meaning?

4. It seems like Figure 7 is arguing that the traditional training recipe (which amounts to setting $\boldsymbol{W} = \boldsymbol{I}$) should not perform well, which is contrary to what we are observing in reality. What is the punchline of this figure, and what am I missing?

5. Why is $\mathrm{SW}_2$ chosen as the performance measure? Is there a particular reason why more “standard” metrics such as FID or the Inception Score were not used?

6. As a follow-up question of the previous one, how exactly was $\mathrm{SW}_2$ calculated?
It is well known that reliably computing $\mathrm{SW}_2$ in high-dimensional spaces is challenging, since two random vectors are almost always nearly orthogonal; as a result, a randomly chosen direction is unlikely to serve as a meaningful proxy for $\boldsymbol{\theta}$ in equation (3), with very high probability. How is this issue addressed?

7. I hope the authors will consider adding visual results for the experiments on CelebA and CIFAR-10 in the paper. It would also be helpful if they could include a brief verbal description of these results in the rebuttal, since presenting figures may be difficult at that stage.

8. For the notion of average geometry to provide meaningful insights into training diffusion models in practice, it seems like it should ideally capture information that evolves as the model parameters (or their distribution) change during training. However, the current definition appears to depend only on the distribution used at initialization, offering no indication of how the geometry of the outputs evolves as the parameters are updated, nor how it would interact with the data distribution during the training. How does the concept of average geometry account for this discrepancy?

---

> ### Author Response · Authors · 2025-11-13
> **Thank you for your review (1/3)**
>
> Thank you for your review. We are pleased that you found our insights novel, our paper timely and that you appreciated the potential impact of our work in serving the greater goal of understanding inductive biases in generative modeling.
>
> We provide answers to your concerns below.
>
> W1. Introduction & SADs definition
> -
>
> As we understand, your main concern is regarding the definition of SADs. **We have now reworded the definition to be more specific regarding "preference"**. To directly answer your question, in this work relative preferences and performance is quantified via the (Max-)Sliced Wasserstein metrics, ($\text{M}$)$\text{SW}_2$, which are properly introduced in Section 3 (see also our answer to your Q5 below).
>
> **Regarding your question about generating data along a a direction, this is literal and accurate**.
>
> Note that we dedicate a large part of Section 3 in understanding diffusion model behavior on rank-one datasets, i.e., $\mathcal{N}(\boldsymbol{0},\boldsymbol{v}\boldsymbol{v}^\top)$ so along the direction $\boldsymbol{v}$, before we generalize our methodology to arbitrary data distributions.
>
> **Of course, if you have any specific suggestions on how to improve the wording please let us know and we will update the paper**.
>
> W2. Related work
> -
>
> **Please note that we have also briefly discussed related work in the background Section 2, right after the introduction**. Related work Section 4.1 is toward the end of the paper and it serves as an extended discussion of our method and findings in relation to prior work. Some of the discussion points are specific and technical, e.g., regarding the average geometry, $\mathbf{G}_{\mathcal{F}}$, and our Conjecture 1 that are introduced in Section 3.2. **For this reason, we believe it is clearer to readers and more appropriate that Section 4.1 remains toward the end of the paper as concluding remarks. However, if you still feel strongly about moving this toward the beginning of the paper please do let us know and we will change this**.
>
> W3. Figures
> -
>
> **In Figure 1** the DiT models are tasked with learning a uniform distribution on a unit sphere in $\mathbb{R}^{256}$. Since a sphere requires three directions to be defined, the figure shows how the choice of three dimensional subspace impacts the resulting learned distribution. On the left we show samples from a model trained on the worst possible choice for the subspace according to our Conjecture 1, i.e., we pick directions corresponding to the largest three eigenvalues of $\mathbf{G}_{\mathcal{F}}$ (or the last three SADs). The right shows the best possible choice according to our conjecture (smallest eigenvalues, first three SADs). The **points clouds are not obscuring anything, they are precisely depicting the learned distributions in each case**. Judging from the resulting simulated "spheres" the right one is of significantly better quality compared to the left although the experimental setup is identical in both cases except for the choice of subspace.
>
>
> **For Figure 2, please also refer to our detailed description provided in the caption and also at the beginning of Section 3**. In particular, here we visualize $\text{MSW}_2$ as a function of the directions $\boldsymbol{v}\in\mathbb{R}^{256}$ drawn from standard bases in signal processing literature. That is, each image is a basis and each pixel corresponds to training a diffusion model on data aligned with a particular basis element. Figure 2 shows two things:
>
> - Harmonic structure, which is hypothesized to be central in convolutional diffusion [1], only loosely characterizes the inductive biases.
>
> - More generally, a random basis is unlikely to be particularly informative. That is, the standard bases we consider do not show clear patterns or trends that reveal the directional inductive biases and, apart from a few distinct observations, performance may be relatively uniform.
>
> In this sense, it motivates our exploration of the average geometry and adaptive bases tailored to the architecture.
>
> **For Figure 3, please also refer to explanations in the beginning of Section 3.1**. In particular, we visualize outputs of the iDDPM architecture when the input is symmetric and all the network weights are also symmetircally initialized. This experiment shows the influence of the resampling layers in the final outputs. Specifically, despite everything being perfectly symmetric, $\texttt{nearest}$ resampling, employed by default in the literature, introduces asymmetry (anisotropy). In this sense, this figure provides a concrete example of a directional bias that is purely due to the architecture and motivates us to uncover architectural biases that may not be as easy to describe (e.g., other directional biases in iDDPM or DiT biases).

---

> ### Author Response · Authors · 2025-11-13
> **Thank you for your review (2/3)**
>
> Q1. Theorem 1
> -
>
> $\lambda_{D-1}\neq\lambda_{D}$ is only necessary to observe anisotropy in deterministic Gradient Descent (GD). **For Stochastic GD (SGD), the phenomenon persists independent of this assumption as we have also remarked right below the theorem**.
>
> **We have now reworded Theorem 1 to be more precise with the mean error and SGD covariance**.
>
> To clarify, this theorem describes optimization dynamics when learning a data distribution of the form $\mathcal{N}(\boldsymbol{0},\boldsymbol{v}\boldsymbol{v}^\top)$ for a unit vector $\boldsymbol{v}$. **The mean (with respect to SGD dynamics and initialization) error is the difference between $\boldsymbol{\Omega}$ and $\boldsymbol{\Omega}_{*}$ (defined in Lemma 1) where $\boldsymbol{\Omega}=\boldsymbol{\Phi}\boldsymbol{\Theta}$ with fixed $\boldsymbol{\Phi}$ (modeling architectural anisotropy) and trainable $\boldsymbol{\Theta}$**. We show that these error signals converge to zero and describe the rate at which this happens as a function of $\boldsymbol{v}$.
>
> **When we write "SGD step with respect to $\boldsymbol{\Theta}$" this means optimizing the score matching objective for the above-described $\boldsymbol{\Omega}$, which only has tunable parameters in $\boldsymbol{\Theta}$ (see above)**. Recall that the theorem applies generally for any eigenvector, $\boldsymbol{v}$, of the matrix $\boldsymbol{\Phi}\boldsymbol{\Phi}^\top$, so **when we write "for $\boldsymbol{v}=\boldsymbol{u}_i$" this is just a particular instantiation of the score matching problems that the theorem is about**.
>
> **Regarding the covariance of SGD steps, this is simply the covariance matrix of the vectorized stochastic gradient with respect to $\boldsymbol{\Theta}$**. The theorem shows that covariance scales with $\lambda_i$ near optimality (where it should ideally be zero). In this sense, the theorem predicts that SGD will be noisier for larger eigenvalues. Hence the claim that large eigenvalue eigenvectors achieve worse performance.
>
> Q2. Markov's inequality
> -
>
> Our central Conjecture 1 is rigorously proven in the linear case (as discussed above). For arbitrary neural networks, **you are correct that the Markov bound is only a heurtistic**: it can be tight for small eigenvalues but not necessarily tight otherwise. **Since we cannot prove things in the general case, we are careful with our labeling and appropriately call our finding a conjecture. Still, beyond the theoretical derivations, we hope that you can appreciate that our conjecture holds under a number of different settings**: it has been validated on synthetic data (rank-one and spheres), image benchmarks (MNIST, CIFAR-10, CelebA-HQ), across different architectures (iDDPM variants and DiT variants) and each experiment has been repeated five times to assess statistical significance.
>
> Q3. Intuition for global geometry / Bimodal distributions
> -
>
> Our definition of geometry is such that we do not artificially inject data-dependent anisotropy, i.e., we focus on biases due to the architecture. As mentioned in Section 3.2, in practice we probe with zeros: this choice avoids injecting data-dependent anisotropy in $\mathbf{G}_{\mathcal{F}}$ and we have found that the results are not sensitive, e.g., we initially also ran some of the experiments of Figure 6 with $\mathcal{N}(\boldsymbol{0},\boldsymbol{I})$ and got similar results. **It not clear how your example of bimodal data challenges our Conjecture 1. The image datasets we have considered are multimodal, e.g., MNIST or CIFAR-10, and yet our conjecture holds even in that case**.
>
> **About robustness of $\mathbf{G}_{\mathcal{F}}(\mathcal{P},\Theta)$ when anisotropy introduced via the probe $\mathcal{P}$: we have now conducted experiments to demonstrate robustness in Appendix F**.
>
> **For some intuition on why it is sensible to expect global structure, note that the aim of the geometry is to isolate architecture-specific biases, not dependent on a particular probe. A concrete and intuitive example of such a bias is given in Section 3.1 and Figure 3 (also see our explanation of this in our answer to your W3)**.
>
>
> Q4. $\boldsymbol{W}=\boldsymbol{I}$ does not perform well
> -
>
> This is correct, the default alignment for the tested datasets appears to be quite significant and our conjecture predicts that this is suboptimal. That is, by a simple linear transformation one achieves substantial improvements for the same training budget, as measured via Wasserstein metrics.
>
> Note that this does not contradict any established works: in practice diffusion models are heavily trained with multi-GPU (or even multi-node) setups, with potentially massive datasets and they further employ tricks like EMA and clipping during inference. **In the limit of infinite training resources and time we do not expect to see any significant differences. What the figure shows is, given a limited training budget, performance is improved via a simple linear transformation, in line with our Conjecture 1**.

---

> ### Author Response · Authors · 2025-11-13
> **Thank you for your review (3/3)**
>
> Q5. Why Wasserstein
> -
>
> Unlike Wasserstein distances, **perceptual "metrics" do *not* correspond to valid statistical distances and can be adversarially manipulated, i.e., good perceptual scores do *not* imply generalization [2, 3]**. In fact, **relying on them is likely to introduce confounders given that the underlying perceptual networks (e.g., Inception) will have their own inductive biases and preferred directions**.
>
> **By definition, perceptual scores are subjective. Our aim is to study optimization dynamics which are inherently objective**.
>
> Q6. How we compute sliced Wasserstein
> -
>
> Please also see Appendix A for details on this and our provided code which details exactly how the Wasserstein metrics are computed. In particular, **when modeling data in $\mathbb{R}^D$ we have found that drawing $L=64D$ random directions is sufficient to capture the sliced metrics to a good-enough accuracy. Note that all of our experiments have been independently repeated five times and we always report averages (and standard errors in Figure 6). We also report the $\text{MSW}_2$, which picks the worst possible slice out of the $L$. In this sense, we are confident that our evaluation is robust and correct**.
>
> We agree that this approach does not scale nicely to higher dimensions but for our experiments, e.g., $D=3\times 32\times 32=3072$ and $L=64\times3072\approx 197k$ this is manageable.
>
>
> Q7. Add visuals
> -
>
> **We have now added the requested visuals in Appendix E**.
>
> As with the MNIST visuals of Figure 7, the default alignment leads to artifacts on the distribution level, i.e., saturated or blank images. However, beyond this observation it is difficult to draw conclusions based on perceptual quality since it is, by definition, subjective and, formally, does not imply an appropriate metric on distributions (see our answer to your Q5 above).
>
> **We caution against the use of visuals in making conclusions; for example, the default alignment may produce more crisp images, which one may prefer. However, this comes at the cost of completely failing to produce anything a large fraction of the time**.
>
> **To give an extreme example of the tension between perception and $\text{SW}_2$, consider a "model" that can only produce a single sample perfectly vs. a model that recovers the overall data distribution but with visible noise**.
>
>
> Q8. Average geometry evolution
> -
>
> As also stressed in answering your Q2, we have experimentally verified that our Conjecture 1 holds across several settings. Therefore, given that our Conjecture 1 holds after thousands of training iterations, **it is reasonable to hypothesize that there exists some kind of invariant in our notion of geometry that persists even during training. In fact, recent work [4], argues that a similar notion of our geometry in discriminative models does indeed posses such invariances**.
>
> **Note that the aim of the average geometry is to capture architecture-specific biases, not dependent on particular weight schemes. A concrete example of such a bias is given in Section 3.1 and Figure 3 (also see our explanation of this in our answer to your W3)**.
>
> ---
>
> **Once again, we would like to thank you for taking the time to review our paper. As most of your concerns were on presentation and clarifications, we hope that our answers and efforts are sufficient for you to consider raising your score and confidence in our work. We remain open to further discussion**.
>
>
> ---
>
> [1] *Generalization in diffusion models arises from geometry-adaptive harmonic representations*, ICLR 2024
>
> [2] *A Note on the Inception Score*, ICMLW 2018
>
> [3] *Exposing flaws of generative model evaluation metrics and their unfair treatment of diffusion models*, NeurIPS 2023
>
> [4] *Geometric Inductive Biases of Deep Networks: The Role of Data and Architecture*, ICLR 2025

---

### Official Review · Reviewer_rSjW · 2025-11-01

**Soundness:** 3
**Presentation:** 3
**Contribution:** 3
**Rating:** 8
**Confidence:** 2

**Summary:**

In this work, the directional preferences of network architecture is theoretically identified and analyzed from the harmonic analysis point of view. The geometry in generative model architecture is clarified and the output space induced by the directional preference of network architecture is decomposed.

**Strengths:**

1. **Clarity**: This work is clearly presented with informative visualization of preliminary results on iDDPM which enhances the motivations. The logic is generally linear and the whole paper is easy to follow. Theories are well articulated and key takeaways are summarized with clarity at the end of each block of analysis and discoveries.

2. **Quality**: The proposed theories are proved comprehensively and coherently on the importance of directional preference in network architectures and the impact of it. The claimed importance of anisotropic conditions is also in general supported by the phenomena in numerical experiments.

**Weaknesses:**

1. Although the claimed discoveries are supported by experiments, datasets used in experiments are in low resolutions where the highest is only $56\times 56$. Experiments on higher dimensionality may indicate results on the contrary to the theoretical discoveries. It would be more comprehensive to conduct experiments on datasets such as ImageNet with the resolution of $256\times 256$.

2. Before this work, there is already existing research discussing and modifying generative models on directional sensitivity, such as Kuramoto Orientation Diffusion Models [1] and Artificial Kuramoto Oscillatory Neurons [2] where [1] proposes a new diffusion model. It would improve the novelty and significance to discuss [1] and [2] and highlight the novelty accordingly.

[1] Yue Song, T. Anderson Keller, Sevan Brodjian, Takeru Miyato, Yisong Yue, Pietro Perona, and Max Welling, Kuramoto Orientation Diffusion Models, 39th Conference on Neural Information Processing Systems, 2025.

[2] Takeru Miyato, Sindy Löwe, Andreas Geiger, Max Welling, Artificial Kuramoto Oscillatory Neurons, ICLR 2025.

**Questions:**

1. Could the authors elaborate on "directional preferences" stated? Are directional preferences equivalent to importance of features in training and inference processes? I am wondering how the directional preferences in network architectures will enhance or reduce the features in data such as images.

2. In terms of the motivation, why is it necessary to study the directional preferences in network architectures for generative models, except its absence but prominence in studies of discriminative models? How does it influence the generation quality of generative models?

3. Could the authors clarify the meaning of $\boldsymbol{v}$? What is the relation between $\boldsymbol{v}$ and $\sigma$?

4. Why is  the best performance achieved when the data is not aligned with the “geometry” that is induced by the score network, but instead lives in the subspace defined by its smallest eigenvalues? Could the authors provide an intuitive explanation?

---

> ### Author Response · Authors · 2025-11-13
> **Thank you for your review (1/2)**
>
> Thank you for your review. We are pleased that you found our presentation clear and praised the quality of our work.
>
> We provide answers to your concerns below.
>
> W1. Experiments in higher dimensions
> -
>
> Unfortunately, due to time and resource constraints, it is unlikely that we will be able to perform such experiments during rebuttal. We elaborate on the fundamental obstacles below:
>
> Note that for a given dimensionality, $D$, unconvering the SADs requires the geometry matrix, $\mathbf{G}_{\mathcal{F}}\in\mathbb{R}^{D\times D}$, which is computed as an expectation over random parameters $\boldsymbol{\theta}$ at initialization. **Taking your example of $D=3\times256\times256\approx197k$ results in a geometry of shape $197k\times 197k$, which is unwieldy and not easily estimated**. In particular, note that higher resolution diffusion typically requires larger networks. This increases the variance of Monte Carlo simulations. That is, we would need to sample and run inference on more and larger networks at initialization to get an accurate estimate of $G_F$. As a separate matter, we note that **ImageNet diffusion typically requires multi-GPU (or even multi-node) setups in the literature, which we cannot replicate given our resources**.
>
> **Another issue is that evaluation of generative models also becomes intractable in higher dimensions**. For example, estimating the $\text{SW}_2$ metrics also requires Monte Carlo simulations that scale poorly with $D$. While the community has gotten around this by use of perceptual metrics like FID or IS, we deem such metrics inappropriate to test our hypothesis. **In particular, unlike $\text{SW}_2$, perceptual "metrics" do *not* correspond to valid statistical distances and can be adversarially manipulated, i.e., good perceptual scores do *not* imply generalization [1, 2]. In fact, relying on them is likely to introduce confounders given that the underlying perceptual networks (e.g., Inception) will have their own inductive biases and preferred directions**.
>
>  Given the above, **we believe that our current experiments, spanning synthetic data, three image benchmarks, convolutional and transformer-based diffusion, each with five independent runs are sufficient to validate our claims. However, we agree (and have explicitly acknowledged this in the limitations Section 4.2 of the paper) that, in general, it is plausible that other explicit or implicit factors could influence or dominate diffusion model optimization dynamics**.
>
> W2. Discussion on Kuramoto works
> -
>
> **We have now cited the works you mentioned in Section 2.1. However, as you acknowledge in your comment, these works are modifying / proposing new diffusion models. In contrast, our work contributes a mathematical framework applicable to arbitrary score-based generative models**. Ours is a theory to explain the directional inductive biases due to architectural geometry, independent from a particular domain or architecture.

---

> ### Author Response · Authors · 2025-11-13
> **Thank you for your review (2/2)**
>
> Q1. Elaborate on direction preferences
> -
>
> **As stated in Section 3.1 of the paper, directional preferences emerge due to architectural choices**. A simple example is given by the iDDPM U-Net in Figure 3, where it is clear that the choice of downsampling layer may bias a network with symmetric weights and a symmetric input to generate asymmetric outputs. In general, however, it is not always possible to cleanly attribute such preferences to specific architectural components (e.g. beyond resampling layers or in DiT) and that is where our proposed $\mathbf{G}_{\mathcal{F}}$ is useful.
>
> These preferences do indeed capture feature importance in the sense that **models will more easily learn data when it is aligned with the SADs (as evidenced in Figures 6 and 8)**.
>
> You then ask about the image domain. **We do validate our theory on image datasets (see Figures 7 and 8). However, note that the focus of our work and framework is not on any particular domain**. Specifically, in our main experiments the geometry is completely decoupled from the data and we therefore expect our findings to be broadly applicable.
>
> Q2. Motivation
> -
>
> **We provide motivation and a broader impact statement in Section 4.3**.
>
> We believe that understanding inductive biases of generative models is an important and largely unsolved problem. This line of research could enable cheaper and more effective architecture design. For example, as evidenced in Figures 6 and 8, our work already shows promise in being able to predict performance prior to training.
>
> As further motivation, we note that by understanding which patterns (i.e., directions ) a model is predisposed to reproduce, this may give us insight into problems such as the memorization of data and hallucinations.
>
> Q3. Clarify meaning of $\boldsymbol{v}$ and $\sigma$
> -
>
> In general, **we use $\boldsymbol{v}$ to denote a unit norm vector in $\mathbb{R}^D$, i.e., a direction**. Please see the beginning of Section 3, where we introduce the problem of learning rank-one distributions with covariance matrix $\propto\boldsymbol{v}\boldsymbol{v}^\top$.
>
> **$\sigma$ means a particular noise level for the standard Gaussian forward process of diffusion models (we standardize notation to variance exploding diffusion)**. This notation is introduced in Section 2.1.
>
> Q4. Intuition on results
> -
>
> Intuitively, we believe our Conjecture 1 holds because the score function models gradient steps. **A direction that is not easily estimated by the score function (i.e., not aligned with its geometry) is one where the reverse diffusion iterates do not change much so data along such directions persists**. In contrast, directions that are well modeled by the score (i.e., aligned with the geometry) will be significantly changed during reverse diffusion and so data along them will get "denoised" heavily.
>
> This intuition is mathematically precise and proven in the linear case (see Section 3.1 and Theorem 1). The case of general networks is analyzed in Section 3.2. In the latter case, we provide intuition and heuristics via a Markov bound which leads to our central Conjecture 1 (experimentally validated in Figures 6 and 8).
>
> ---
>
> **Once again, we would like to thank you for taking the time to review our paper. We hope our answers have increased your confidence in our work. We remain open to further discussion**.
>
> ---
>
> [1] *A Note on the Inception Score*, ICMLW 2018
>
> [2] *Exposing flaws of generative model evaluation metrics and their unfair treatment of diffusion models*, NeurIPS 2023

---

### Comment · Area_Chair_QzKi · 2025-11-26
**Reminder: Please review and follow up on author responses**

Dear reviewers,

This is a gentle reminder that the authors have posted their responses. If you have any questions or need further clarification, please feel free to share your follow-up comments.

Thank you very much for your time—your contribution is greatly appreciated and valuable to the community.

Best,

Area Chiar

---

### Meta-Review · Area_Chair_NKW7 · 2026-01-07

**Summary:**

This paper studies the inductive biases induced by network architectures in diffusion models. They construct a matrix, obtained by computing the outer product of the network against itself, averaged over initializations and a probing distribution over inputs. They claim that the eigenvalues of this matrix contain information about these inductive biases, which they prove to be the case in a simplified setting with Gaussian data on a 1-dimensional subspace and a linear model. Unfortunately, I believe the authors' rebuttal does not fully address relevant concerns from the reviewers (details below), and I thus recommend rejection.

**Reviewer Concerns:**

The reviewers had several concerns, most notably:

- The scalability of the method being poor, which the authors conceded.

- The use of the sliced Wasserstein distance as a metric of generative model performance. The authors mentioned in their rebuttal wanting to avoid using common metrics which rely on pre-trained networks (like the inception network or DINO) to avoid the inductive biases of these networks, and because computing Frechet distance on these spaces does not produce a mathematical distance between distributions. I believe the author's argument to be overly dismissive of the reviewers' concern: using a Wasserstein distance will also have some biases due to the choice of distance function (e.g. why would using Euclidean distance for the Wasserstein distance be appropriate for images?).

- The theoretical setting is overly simplified. While the authors provided some intuitions as to why they believe their intuitions carry over to more complex settings, I still believe that this is a valid concern about the paper.

- Several minor misunderstandings, that I believe were successfully clarified by the authors in their rebuttal.

**Reviewer Scores:**

I do not believe the authors would have raised their score.

---

### Decision · Program_Chairs · 2026-01-26

Reject